# Development and characterization of functional antibodies targeting NMDA receptors

Nami Tajima[1,3], Noriko Simorowski[1,3], Remy A. Yovanno [2,3], Michael C. Regan[1], Kevin Michalski[1], Ricardo Gómez [1], Albert Y. Lau [2✉] & Hiro Furukawa [1✉]

*N*-methyl-D-aspartate receptors (NMDARs) are critically involved in basic brain functions and neurodegeneration as well as tumor invasiveness. Targeting specific subtypes of NMDARs with distinct activities has been considered an effective therapeutic strategy for neurological disorders and diseases. However, complete elimination of off-target effects of small chemical compounds has been challenging and thus, there is a need to explore alternative strategies for targeting NMDAR subtypes. Here we report identification of a functional antibody that specifically targets the GluN1-GluN2B NMDAR subtype and allosterically down-regulates ion channel activity as assessed by electrophysiology. Through biochemical analysis, x-ray crystallography, single-particle electron cryomicroscopy, and molecular dynamics simulations, we show that this inhibitory antibody recognizes the amino terminal domain of the GluN2B subunit and increases the population of the non-active conformational state. The current study demonstrates that antibodies may serve as specific reagents to regulate NMDAR functions for basic research and therapeutic objectives.

[1] W.M. Keck Structural Biology Laboratory, Cold Spring Harbor Laboratory, Cold Spring Harbor, New York, NY 11724, USA. [2] Department of Biophysics and Biophysical Chemistry, Johns Hopkins University School of Medicine, 725 N. Wolfe Street, WBSB 706, Baltimore, MD 21205, USA. [3] These authors contributed equally: Nami Tajima, Noriko Simorowski, Remy A. Yovanno. ✉email: alau@jhmi.edu; furukawa@cshl.edu

N-methyl-D-aspartate receptors (NMDARs) belong to the family of ionotropic glutamate receptors (iGluRs) that are involved in the majority of fast excitatory neurotransmission. NMDARs are mostly expressed in the central nervous system[1], but recent studies demonstrate that they are also expressed in tumors and that the activity of NMDARs controls their invasiveness[2,3]. NMDARs form heterotetrameric ion channels composed of the obligatory GluN1 subunits and GluN2 (A-D) and/or GluN3 (A-B) subunits[1,4–6]. The GluN1 and GluN3 subunits bind glycine or D-serine, whereas the GluN2 subunits bind the excitatory neurotransmitter glutamate. All of the subunits contain an amino-terminal domain (ATD), a ligand-binding domain (LBD), a transmembrane domain (TMD), and a carboxyl-terminal domain (CTD), which interact with each other in defined manners to mediate functions including channel gating, allosteric modulation, and cellular signaling[4,7–10].

There have been a number of high-resolution x-ray crystallographic structures of fragmented extracellular domains that show binding modes of compounds and ions to the LBDs[8,11–21] and ATDs[7,22–27]. More recently, a number of studies on intact tetrameric NMDARs showed that the subunits are arranged as a dimer of GluN1-GluN2 heterodimers and that domains and subunits move in discrete patterns to control channel gating and allosteric modulation[7–9,28–34].

An important hallmark of NMDARs is the subtype diversity created by different combinations of the subunits above, which result in the formation of receptor ion channels with different compound binding profiles, speeds of activation, deactivation, desensitization, and spatio-temporal expression patterns[35]. Subtype-specific targeting of NMDARs has been vigorously pursued over the past two decades for their promise in therapeutic interventions for various neurological diseases and disorders, and possibly for cancer. Thus far, efforts to target NMDAR subtypes rely exclusively on small molecules, however, the majority of the compounds have not reached clinical usage except for memantine and ketamine due mainly to side effects including hallucination, motor dysfunction, and memory loss, which are likely caused by non-specific off-target binding[36,37]. Antibody-based therapeutic approaches have been enthusiastically pursued over many years with the prime example of successful cases being ant-programmed cell death protein 1 (PD1) and anti- checkpoints T-lymphocyte-associated protein 4 (CTLA-4) cancer immune-therapies[38]. There are fewer antibody-based therapies for neurological diseases compared to other diseases caused by deficits in peripheral tissues and organs at this point. However, an anti-amyloid antibody[39] that targets beta-amyloid in the brain poses an intriguing possibility for delaying the age-of-onset of Alzheimer's disease.

Here we explore the possibility of subtype-specific targeting and regulation of the GluN1-GluN2B NMDAR by antibodies. We report that an antibody against the GluN1-GluN2B NMDAR can specifically downregulate channel functions by binding to the ATD and stabilizing the receptors in the non-active conformation. The current study opens a unique avenue for regulating NMDAR channels via antibodies.

## Results

**Identification and characterization of anti-GluN1-GluN2B NMDAR inhibitory antibodies**. To isolate functional antibodies against the GluN1-GluN2B NMDARs, we immunized mice with purified intact rat GluN1a-GluN2B NMDAR proteins prepared in lauryl maltose neopentyl glycol (LMNG)[40]. We isolated ~30 monoclonal antibodies (mAbs) from a mouse with no apparent neuropsychiatric consequences (see Methods) and specifically selected for ones that recognize folded regions of the

NMDAR protein rather than flexible loops or denatured proteins. Such 'folding-specific' antibodies typically recognize the protein surface and have a higher tendency to alter functions of target proteins as demonstrated previously[41]. Toward this end, we screened for IgGs that showed signal in an enzyme-linked immunosorbent assay (ELISA) using the intact rat GluN1a-GluN2B NMDAR proteins in the presence of 0.01% LMNG and no signal in Western blotting executed in a denaturing condition (Fig. 1a). We identified four antibodies that satisfied the above criteria and found one of them, IgG2, that inhibits the activity of the GluN1-1b (hence GluN1b)-GluN2B NMDARs as measured by two-electrode voltage clamp (TEVC) on cRNA injected *Xenopus laevis* oocytes (Fig. 1b). The inhibition occurs in a concentration-dependent manner (Fig. 1b, f). Importantly, little or no effect was observed when IgG2 was applied to the oocytes expressing the GluN1b-GluN2A, GluN1b-GluN2C, and GluN1b-GluN2D NMDARs, indicating that this inhibitory effect is specific to the GluN1b-GluN2B NMDARs (Fig. 1c–e). Another 'protein folding-specific' antibody, IgG5, has a minor potentiating effect rather than an inhibitory effect, implying that the approach to control NMDAR functions by antibodies may be applicable to both upregulation and downregulation (Fig. 1g).

Next, we tested if the variable fragment (Fv) of IgG2 (Fv2) is capable of inhibiting the activity of the GluN1b-GluN2B NMDAR. Toward this end, we cloned cDNA of heavy and light chains of the Fv2 from the hybridoma cell line that expresses IgG2, recombinantly expressed them in *Brevibacillus choshinensis*, and purified the assembled Fv2 to homogeneity (Supplementary Fig. 1a–c and Methods). The purified Fv2 fragment is capable of binding specifically to the GluN1-GluN2B NMDA receptor as assessed by peak shifts in fluorescence-coupled size exclusion chromatography (FSEC) using intrinsic tryptophan fluorescence (Excitation/Emission = 280/330 nm) (Supplementary Fig. 1d, e). Also, the Fv2 fragment is able to inhibit the GluN1b-GluN2B NMDAR current (~60.0% of maximum current at 0.1 mg/ml) indicating that the critical factor for inhibition is binding of Fv but not cross-linking by IgG (Fig. 1h).

Furthermore, we tested if the inhibitory effect of IgG2 is independent of expression hosts. For this, IgG2 and the antigen-binding fragment (Fab) of IgG2 (Fab2) were tested on the GluN1b-GluN2B NMDARs expressed on HEK293 cells (Supplementary Fig. 2). We observed a similar inhibition pattern in whole-cell patch-clamp recordings by both IgG2 and Fab2 to the one detected in *Xenopus* oocytes. In the patch-clamp experiments, the inhibition reached the maximum within five seconds of IgG2 or Fab2 application, which is likely faster than the process of receptor internalization.

Lastly, we tested the effect of IgG2 on the GluN1-1a splice variant which does not contain the exon 5-encoded motif in ATD and the GluN1-1a-GluN2A-GluN2B tri-heteromeric NMDARs in HEK293 cells. The GluN1-1a-GluN2B NMDAR showed inhibition by IgG2 (Supplementary Fig. 2) indicating that the alternative splicing does not affect the inhibition. The GluN1-1a-GluN2A-GluN2B NMDAR showed a decreased level of inhibition compared to the GluN1-1a-GluN2B NMDAR indicating that the number of antibody binding per tetrameric channel controls the extent of inhibition (Supplementary Fig. 2). Furthermore, this set of experiments showed that the application of IgG2 elicited a decrease in peak current, an increase in the extent of desensitization, and a faster speed of desensitization in both GluN1-1a-GluN2B and GluN1-1a-GluN2A-GluN2B NMDARs.

**Isolated GluN1b-GluN2B ATD recognizes functional antibodies**. We next attempted to identify the domain within the NMDARs responsible for binding to IgG2 and IgG5. Toward this

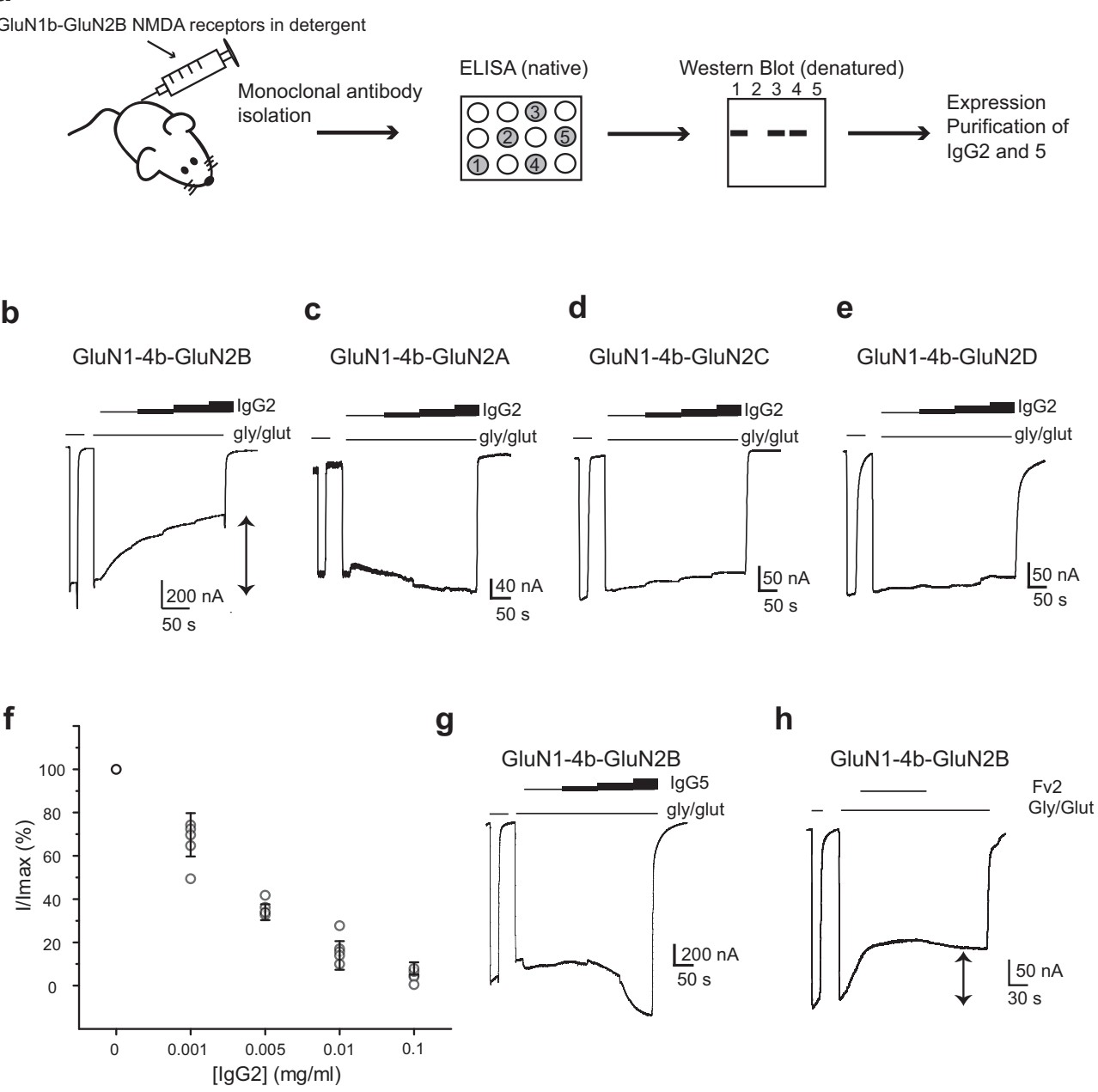

**Fig. 1 Isolation and characterization of anti-GluN1-GluN2B NMDA receptor IgGs. a** Monoclonal antibodies were produced by mouse-immunization by intact rat GluN1b-GluN2B NMDA receptors purified in LMNG. Clones that produced signal in ELISA and no signal in Western blot were isolated. **b–f** Glycine/Glutamate-evoked currents measured by TEVC on cRNA injected *Xenopus* oocytes expressing rat GluN1b-2B, GluN1b-2A, GluN1b-2C, and GluN1b-2D in the presence of various concentrations (0.001–0.1 mg/ml) of purified IgG2. The specific inhibitory effect of IgG2 on the GluN1b-GluN2B NMDA receptors are dose dependent. Symbols and error bars in panel f represent mean ± SD for five independent recordings from five different oocytes. **g** Application of various concentrations (0.001–0.1 mg/ml) of IgG5 has no inhibitory effect but has a slight potentiating effect at 0.1 mg/ml (111 ± 7.5% - mean ± SD; $n = 5$ where $n$ is the number of oocytes used for independent recordings). **h** The Fv fragment of IgG2 (Fv2) retains an inhibitory capability. Shown here is the current recorded in the presence of 0.1 mg/ml of Fv2.

end, we tested interactions between the isolated GluN1b-GluN2B ATD proteins[22,24,25] and IgG2, IgG5, or Fab fragments of IgG2 and IgG5 (Fab2 and Fab5; see Methods) by FSEC using intrinsic tryptophan fluorescence (Excitation/Emission = 280/330 nm) as the detection method (Fig. 2). In these experiments, peak shifts (~200 sec) in FSEC were observed between GluN1b-GluN2B ATD and GluN1b-GluN2B ATD mixed with IgG2 (Fig. 2a) or Fab2 (Fig. 2b), indicating binding. No such shift was observed when GluN1b-GluN2A ATD was mixed with IgG2 or Fab2

confirming subtype-specific binding (Fig. 2c, d). A similar peak shift pattern was observed for IgG5 and Fab5 when mixed with GluN1b-GluN2B ATD but not with GluN1b-GluN2A ATD (Fig. 2e–h). Overall, the FSEC experiments above indicated that the GluN2B ATD alone may participate in the binding of IgG2, Fab2, IgG5, and Fab5. A remaining possibility that the antibodies might interact partly with other domains including LBDs and TMDs was eliminated by subsequent structural biological studies described in the next sections.

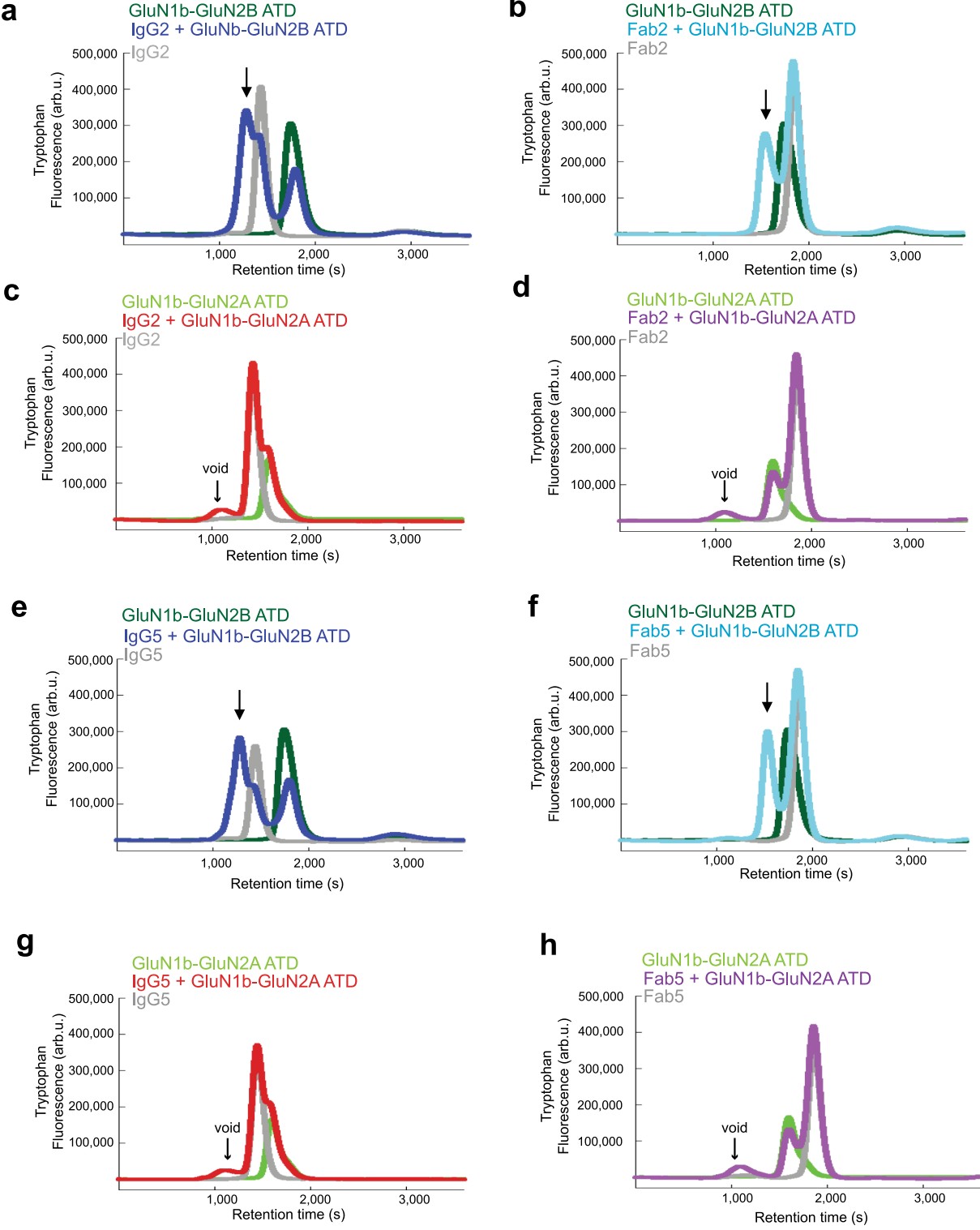

**Fig. 2 Subtype-specific binding of anti-GluN1-GluN2B NMDA receptor antibodies. a–d** Purified IgG2 (panels **a** and **c**) or Fab2 (panels b and d) are mixed with GluN1b-GluN2B ATD (panels **a–b**) or GluN1b-GluN2A ATD (panels **c–d**) heterodimeric proteins and subjected to Superdex200 size-exclusion chromatography using tryptophan fluorescence (280 nm/330 nm = excitation/emission) as a detection method. Arrows indicate shifted peaks compared to non-mixed controls. **e–h** Equivalent experiments for IgG5 (panels **e** and **g**) and Fab5 (panels **f** and **h**) where GluN1b-GluN2B ATD (panels **e–f**) and GluN1b-GluN2A ATD (panels **g–h**) were mixed. The color code for chromatographs is shown on top of in each panel.

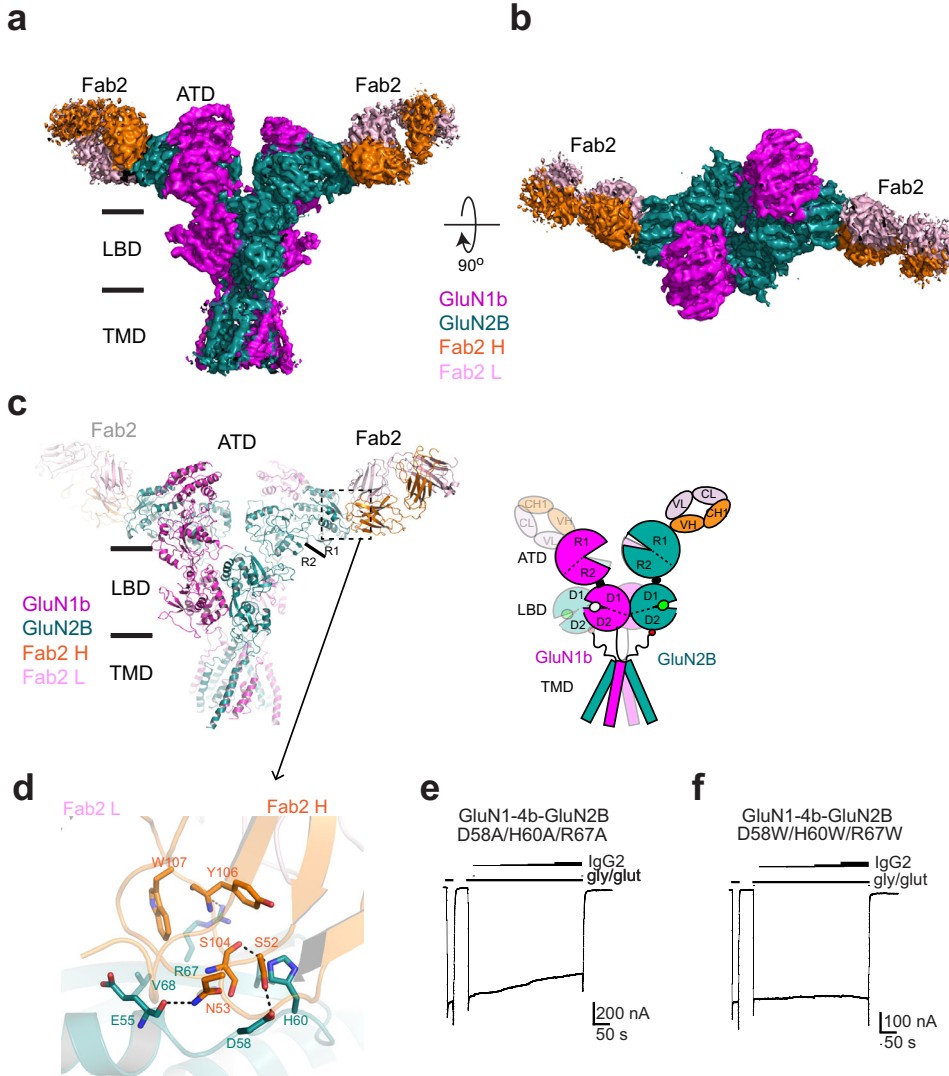

**Fig. 3 Structural analysis of agonists-bound GluN1b-GluN2B NMDARs complexed to Fab2. a–b** Cryo-EM density of the non-active1 3D class at overall resolution of 3.92 Å from the 'side' (**a**) and 'top' (**b**) of the N-terminus. Densities for TMD, LBD, and ATD of GluN1b (magenta) and GluN2B (dark green) and Fab2 heavy chain (orange) and light chain (light pink) were observed. The structure was solved in the presence of 1 mM glycine and glutamate. **c** The molecular model built based on the cryo-EM density in the same color code as in panels **a** and **b**. **d** Zoomed view of the ATD-Fab2 interaction site demonstrating residues from CDR 1 and 3 of the heavy chain are mediating polar and hydrophobic interactions. **e–f** Interacting residues, Asp58, His60, and Arg67, were mutated to Ala (panel **e**) or Trp (panel **f**) and measured for inhibitory effects as in Fig. 1. The Ala triple mutant slightly retains the inhibitory effect whereas the Trp triple mutant completely removes the effect.

**Single-particle cryo-EM on GluN1b-GluN2B NMDAR–Fab2 complex.** We next sought to identify the binding site of Fab2 within the GluN1b-GluN2B NMDARs in order to understand the potential mechanism of inhibition by Fab2. Toward this end, we purified the intact GluN1b-GluN2B NMDAR proteins[7,8] in the presence of 1 mM glycine and 1 mM glutamate and complexed them with the purified Fab2 fragment (see Methods). We subjected the GluN1b-GluN2B NMDAR-Fab2 complex, but not the GluN1b-GluN2B NMDAR-IgG2 complex, to cryo-EM since the former showed higher sample homogeneity, which resulted in electron micrographs with evenly dispersed particle distribution (Supplementary Fig. 3a). Single-particle analysis resulted in three 3D classes that differed from each other mainly in their ATD conformations (Fig. 3, Supplementary Fig. 3). The resolution of the 3D reconstructions ranged from 3.9 to 6.6 Å as estimated by Fourier shell correlation (FSC) curves (Supplementary Fig. 3b and Supplementary Table 1), thus, the quality of the cryo-EM density maps was mostly sufficient to identify and trace

secondary structural elements as well as the side chains of bulky residues (Supplementary Fig. 3d). Furthermore, we solved an x-ray crystallographic structure of Fab2 at 2.5 Å to facilitate model building into the cryo-EM density (Supplementary Fig. 4 and Supplementary Table 2).

The cryo-EM structures unambiguously showed that Fab2 binds to the R1 lobe of the GluN2B ATD (Fig. 3c). One heterotetrameric GluN1b-GluN2B NMDAR channel is capable of binding two Fab2 fragments at the equivalent region of the two GluN2B subunits. The single-particle cryo-EM showed density for the ATD, the LBD, and the TMD of the GluN1b-GluN2B NMDARs along with the density for the Fv portion of the Fab2 fragment, which interacts only with the GluN2B ATD. The majority of antibody binding involves the GluN2B residues Ser31, Glu55, Asp57-58, Phe59, His60, and Arg67 within the ATD and residues from complementary determining region (CDR) 2 and CDR 3 from the heavy chain of IgG2 (Fig. 3c, d). The GluN2B residues involved in IgG2 binding are not conserved in the other

GluN2 subtypes (A, C, and D), consistent with the subtype-specific inhibitory effect shown in the electrophysiological experiments (Fig. 1). The binding residues are conserved among the mammalian GluN2B subunits, implying that the IgG2 will likely recognize and inhibit GluN1-GluN2B NMDARs from other mammalian species.

**Inhibition by Fab2 is mediated by direct binding onto the GluN2B subunit.** To understand whether the inhibitory effect was due to specific binding of the IgG2 or Fab2 to the GluN2B ATD as observed in the cryo-EM structure or other factors, we conducted site-directed mutagenesis on the interacting residues on the GluN2B subunit and tested the inhibitory effect on the ion channel activity (Fig. 3e, f). We focused on the three residues, Asp58, His60, and Arg67 from GluN2B, whose side-chain atoms, not main chain atoms, are involved in binding, and therefore, are amenable to mutagenesis (Fig. 3d). Specifically, we mutated these residues to alanine or tryptophan. These mutant receptors did not show detectable binding when assessed by peak shifts in FSEC (Supplementary Fig. 5). In the alanine mutant, inhibition of IgG2 was mostly but not completely removed (Fig. 3e), perhaps indicating that the alanine mutations did not completely mask the binding capability of the other four binding residues (i.e., main-chain atoms of Ser31, Glu55, Asp57, and Phe59). This plausible weak binding was not detected by the FSEC analysis. The tryptophan mutant that sterically rules out the GluN2B ATD – IgG2 interaction showed complete abolishment of the inhibition by IgG2 (Fig. 3f). Overall, the above results confirm that the inhibition is mediated by a direct interaction between the GluN2B ATD and IgG2.

**Fab5 and Fab2 bind distinct surfaces of GluN1 and GluN2B ATDs.** To understand the underlying factors that contribute to the different effects exhibited by IgG2 and IgG5, we sought to determine the binding site for IgG5. For this, we implemented single-particle cryo-EM on the intact GluN1b-GluN2B NMDAR-Fab5 complex (Fig. 4) and x-ray crystallography on the GluN1b-GluN2B ATD complexed to Fab5 (Supplementary Fig. 6). Together, these two structural analyses delineated the binding sites as well as protein conformational states in the context of the intact NMDAR channel.

The cryo-EM structure of GluN1b-GluN2B NMDAR-Fab5 was obtained at resolutions ranging from 4.45 to 7.51 Å as estimated by Fourier shell correlation (FSC) curves (Supplementary Fig. 7 and Supplementary Table 3). Although the overall quality of the cryo-EM density is inferior to that of GluN1b-GluN2B NMDAR-Fab2, it is sufficient to capture patterns of conformational alteration. The specific residues involved in the interaction between GluN1b-GluN2B NMDAR and Fab5 were captured by the x-ray crystallographic structure of GluN1b-GluN2B ATD-Fab5 at 4.54 Å (Supplementary Fig. 6 and Supplementary Table 1), which shows that the binding involves His311, Ser312, Phe313, Gln331, Ser332, Asn333, and Met334 of GluN2B in the R1 lobe and residues from CDR1 and CDR3 of the light and the heavy chains of IgG5, respectively. Although both IgG2 and IgG5 bind the GluN2B ATD, they do so at distinct locations (Fig. 4e, f).

**Fab2 favors the nonactive conformation of GluN1b-GluN2B NMDAR.** Extensive 3D classification in the single-particle analyses revealed discrete conformations that were also observed in our recent studies on GluN1b-GluN2B NMDARs with no bound antibodies[7,8]. In these studies, GluN1b-GluN2B NMDARs in the presence of glycine and glutamate reside in the three major conformations, nonactive1, nonactive2, and active. Nonactive1 and nonactive2 contain closed and open GluN2B ATD bi-lobes,

respectively. In the active conformation, the GluN2B ATD bi-lobe is open, and the heterodimeric interface of the GluN1b-GluN2B ATD is rearranged, which results in a rolling motion of the two GluN1b-GluN2B LBD heterodimers to open the channel gate (Fig. 5). Thus, the prerequisite for activation of the GluN1b-GluN2B NMDAR is opening of the GluN2B ATD bi-lobe. On the other hand, stabilization of the closed GluN2B ATD favors inhibition. Non-active1 is similar to the conformation of the receptor bound to an allosteric inhibitor such as ifenprodil that stabilizes the closed GluN2B ATD bi-lobe[22,24].

The cryo-EM data for the GluN1b-GluN2B NMDAR-Fab5 in the presence of glycine and glutamate was classified into four 3D classes (Fig. 5), where one corresponds to active, another is similar to nonactive1, and the other two are similar to nonactive2. This conformational pattern is similar to that of the GluN1b-GluN2B NMDAR with no antibodies bound[7,8], thus, is consistent with the observation that IgG5 does not mediate inhibition and instead has a small potentiating effect on the function of GluN1b-GluN2B NMDARs.

The cryo-EM data for the GluN1b-GluN2B NMDAR-Fab2 in the presence of glycine and glutamate was classified into three similar 3D classes, where two of them correspond to nonactive1 with the closed GluN2B ATD bi-lobe, and the other corresponds to nonactive2-like but with the GluN2B ATD bi-lobe only slightly open (Fig. 6). The closed GluN2B ATD bi-lobe disallows sufficient reorientation of the GluN1b-GluN2B ATD interface to cause rolling of the GluN1b-GluN2B LBD heterodimers, thus, the channel gate is closed. There is no clear evidence for the presence of protein conformations representing the active and non-active2 conformations with wide-open GluN2B ATD bi-lobes. Therefore, we suggest that the mechanism of inhibition by Fab2 or IgG2 may involve an alteration of the free energy landscape that results in unfavorable transitions from nonactive1 to nonactive2 and active conformations by stabilization of the closed GluN2B ATD bi-lobe.

**Molecular dynamics simulations of antibody-ATD interactions.** Allosteric inhibitors such as zinc stabilize closed GluN2A and GluN2B ATD bi-lobes by binding at the inter-R1-R2 cleft and tethering residues from the R1 and R2 lobes[9,23–25]. Our cryo-EM structures show that Fab2 binding favors GluN2B bi-lobe closure and stabilization of the non-active1 conformation, even though the binding site is at the 'top' of the GluN2B R1 lobe and not at the inter-R1-R2 cleft. Thus, GluN2B ATD bi-lobe closure induced by Fab2 is mediated by a different mechanism, likely involving long-range interactions. We probed such interactions using all-atom molecular dynamics (MD) simulations of either Fv2 or Fv5 bound to the extracellular GluN1b-GluN2B tetramer (ATD-LBD), and applied dynamical network analysis to the resulting trajectories and compared the results. Specifically, we quantified interaction strength resulting from coupled pairwise residue motions by computing generalized correlation coefficients[42]. This procedure allows us to determine how the Fv2 fragment dynamically alters the strength of interactions formed with the NMDAR ATDs and stabilizes a network of interactions at relevant NMDAR subunit/domain interfaces to mediate allosteric inhibition. Here we focused on the Fv fragments since they mediate similar functional effects to Fabs.

First, application of our dynamical network model to ATD-Fv2 and ATD-Fv5 shows that there are stronger and more localized interactions that persist over time between Fv2 and the ATD than between Fv5 and the ATD (Fig. 7a, b). For Fv2, the most strongly correlated interface interactions ($r_{MI} > 0.3$ for ≥15/20 windows) exist between the R1 lobe of the GluN2B ATD (GluN2B ATD R1) and the heavy chain CDR loops of Fv2, especially around the H3

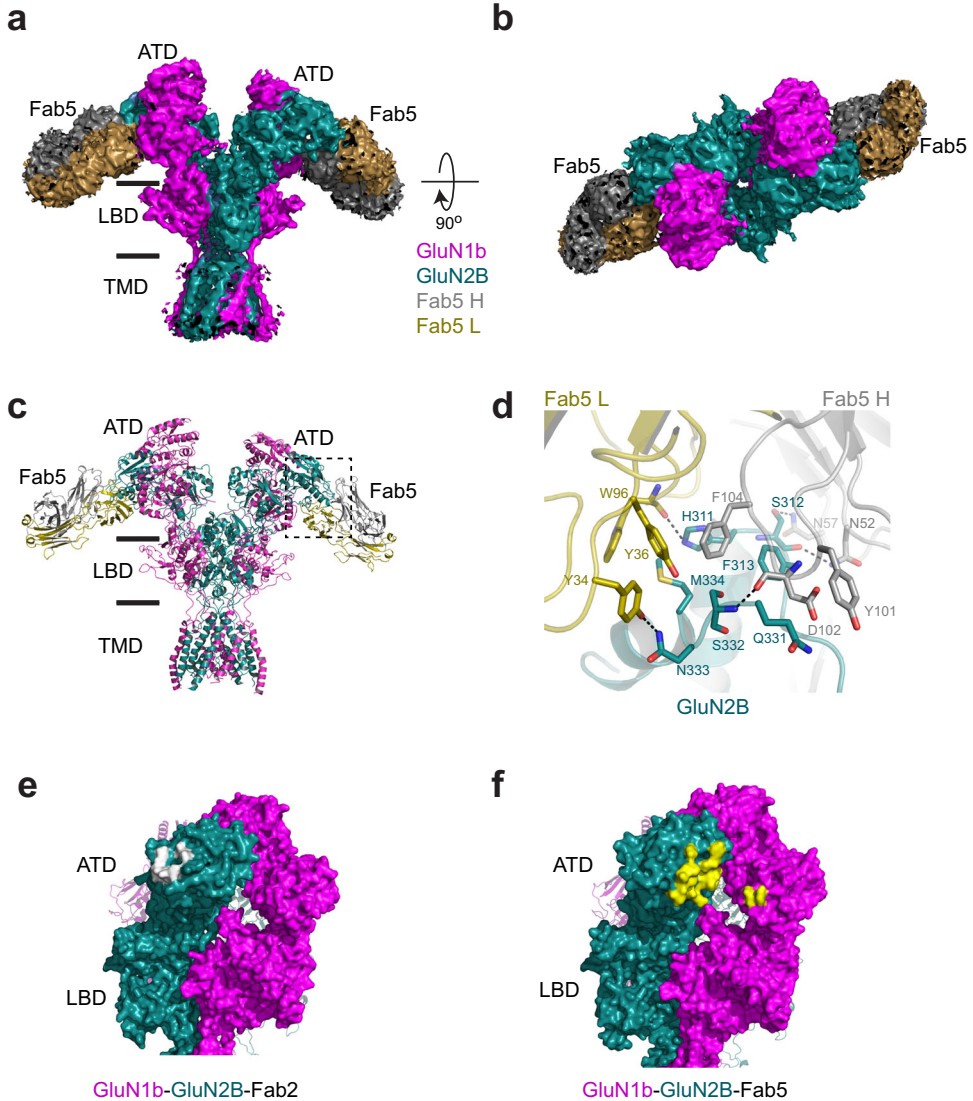

**Fig. 4 Structural analysis of agonists-bound GluN1b-GluN2B NMDARs complexed to Fab5. a–b** Cryo-EM density of non-active2-like 3D class at overall resolution of 4.45 Å from the 'side' (panel **a**) and 'top' (panel **b**) of the N-terminus. Densities for TMD, LBD, and ATD of GluN1b (magenta) and GluN2B (dark green) and Fab5 heavy chain (gray) and light chain (dark gold) were observed. The structure was solved in the presence of 1 mM glycine and glutamate. **c** The molecular model built based on the cryo-EM density in the same color code as in panels **a** and **b**. **d** Zoomed view of the ATD-Fab5 interaction site. Shown here is the crystal structure of the GluN1b-GluN2B ATD complexed to Fab5 at 4.54 Å. The molecular model around the GluN2B-Fab5 interface fits well into the cryo-EM density. Binding of Fab5 involves residues from CDR 1 and 3 of the light chain and CDR 2 and 3 of the heavy chain. **e–f** Surface presentation of residues interacting with Fab2 (panel **e**, white surface) and Fab5 (panel **f**, yellow surface) illustrating that there is no overlap between the binding sites.

loop. Additionally, the L1 and L3 loops of Fv2 interact with the GluN2B ATD R1 (Fig. 7a). Our simulations of Fv5-ATD show that Fv5 interacts with both GluN1b ATD R2 and GluN2B ATD R1, although with fewer correlated interface interactions (Fig. 7b). Furthermore, mean shift clustering analyses[43] reveal structurally compact binding modes for Fv2, whereas Fv5 exhibits a diverse set of binding modes, implying that the more defined binding of Fv2 compared with Fv5 may account for a more robust functional effect of Fv2 (Supplementary Fig. 8). A possible explanation for the structural flexibility of Fv5 is its increased protein contact surface area (11.82 nm$^2$) compared to Fv2 (9.86 nm$^2$) coupled with more spatially distributed contact residues along the ATDs[44].

We next analyzed the GluN1b-GluN2B subunit interface within the ATD heterodimer. The two GluN1b-GluN2B ATD heterodimers in the tetramer have similar trajectories, with the

exception of premature partial closure of the bi-lobe occurring in the C/D subunit of Fab5. For that reason, we focus our analysis on the A/B heterodimer. In both Fv2-ATD and Fv5-ATD, there are extensive interfaces between the R1 lobes of the GluN1b and GluN2B ATDs via interactions between the following regions: α2 of GluN1b with α1', α2', and the hypervariable loop (HVL') of GluN2B; α3 of GluN1b with α1' and α2' of GluN2B; and the HVL of GluN1b with α1' and α2' of GluN2B[24] (Fig. 7c, d). The major difference is that the Fv2-ATD has a greater number of persistently correlated interactions at the interface between GluN1b ATD R1 and the GluN2B ATD R2 (Fig. 7c; gray oval), which are mostly absent in our simulations of Fv5-ATD (Fig. 7d). These GluN1b ATD R1-GluN2B ATD R2 interactions stabilize the closed bi-lobe of the GluN2B ATD as observed in our cryo-EM structures of GluN1b-GluN2B NMDARs in the nonactive1 conformation. Another difference is that the R2 lobes of the two

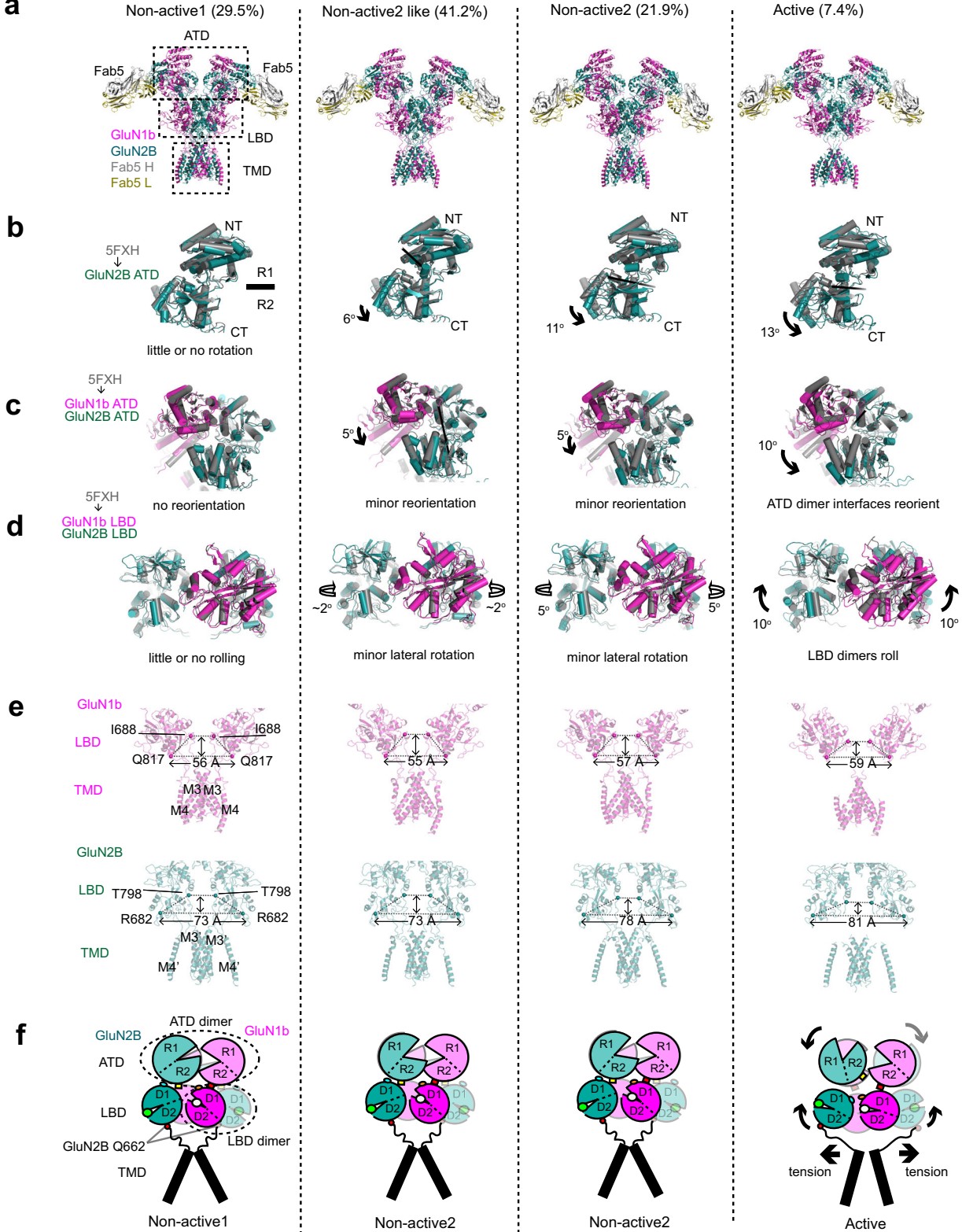

**Fig. 5 Conformational states of GluN1b-GluN2B NMDAR-Fab5 complex. a** 3D classes of the GluN1b-GluN2B NMDAR-Fab5 in the presence of glycine and glutamate. The four 3D classes belong to nonactive1, nonactive2-like, nonactive2, and active conformations. **b**–**d** The structures of GluN2B ATD (panel **b**), GluN1b-GluN2B ATD heterodimer (panel **c**), and GluN1b-GluN2B LBD heterodimers (panel **d**) from GluN1b-GluN2B NMDAR-Fab5 are compared to those of GluN1b-GluN2B NMDAR in nonactive1 (PDB code 5FXH; colored gray). **e** Side views of GluN1b (magenta, upper panel) and GluN2B (dark green, lower panel) showing residues around the channel gating ring (GluN1b Gln817 and GluN2B Arg682 in spheres). **f** Schematic presentation of interdomain and –subunit movements.

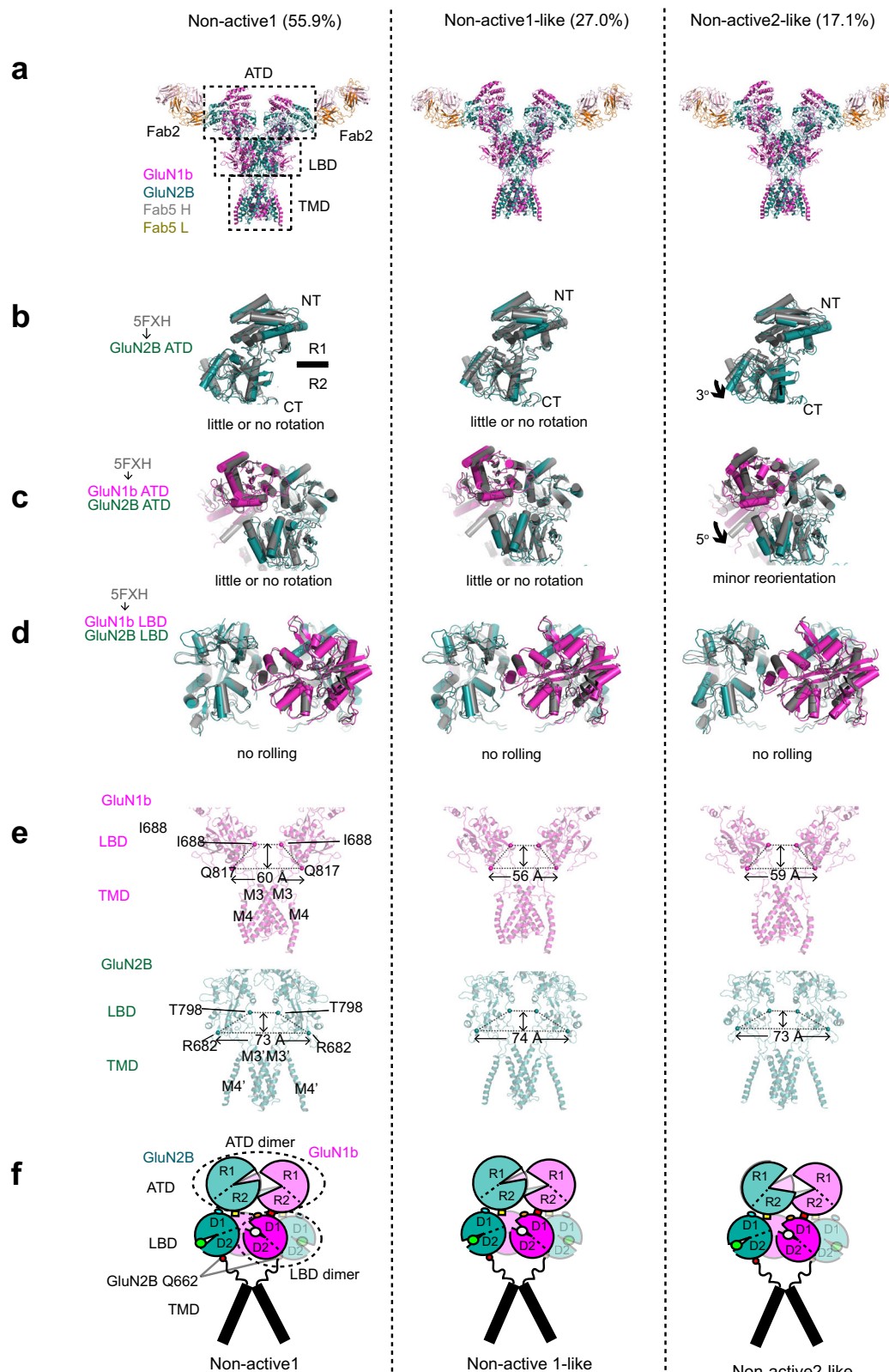

**Fig. 6 Conformational states of GluN1b-GluN2B NMDAR-Fab2 complex. a** 3D classes of the GluN1b-GluN2B NMDAR-Fab2 in the presence of glycine and glutamate. The four 3D classes belong to nonactive1, nonactive2-like, non-active2, and active conformations. **b**–**d** The structures of GluN2B ATD (panel **b**), GluN1b-GluN2B ATD heterodimer (panel **c**), and GluN1b-GluN2B LBD heterodimers (panel **d**) from GluN1b-GluN2B NMDAR-Fab5 are compared to those of GluN1b-GluN2B NMDAR in nonactive1 (PDB code 5FXH; colored gray). **e** Side views of GluN1b (magenta, upper panel) and GluN2B (dark green, lower panel) showing residues around the channel gating ring (GluN1b Gln817 and GluN2B Arg682 in spheres). **f** Schematic presentation of inter-domain and –subunit movements.

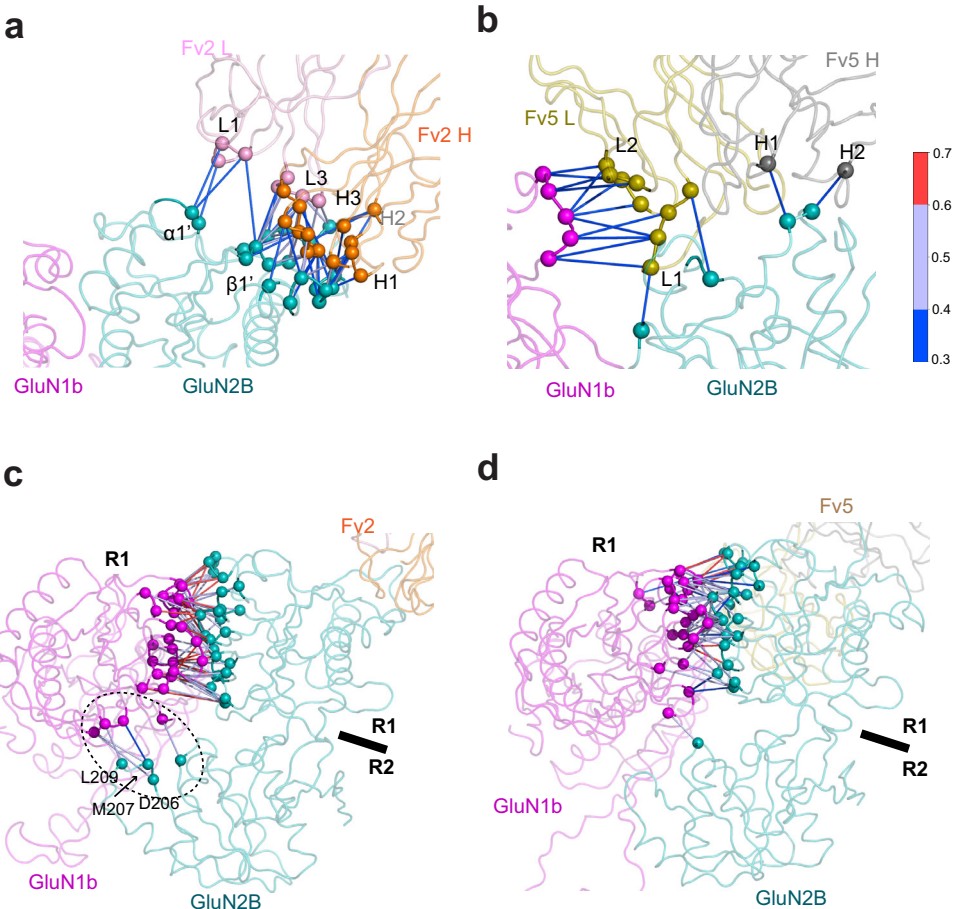

**Fig. 7 MD simulation of Fv interactions with the GluN1b-GluN2B ATD dimer. a–b** A dynamical network representation of the interface contacts between Fv2 and the GluN2B ATD R1 lobe (panel **a**) and Fv5 and the GluN1b R2 and GluN2B ATD R1 lobes (panel **b**). The spheres correspond to Cα atoms, or nodes in the network, pairs of which are connected by a line (or edge) if the heavy atoms of their residues are within 6 Å for at least 75% of the trajectory. To highlight the most important interactions, we only show contact pairs present in at least 15/20 simulation windows and exhibit a correlation coefficient greater than 0.3. These selection criteria are used in all subsequent network representations unless indicated otherwise. The specific value of the correlation coefficient between each pair is indicated by the color of the line connecting them. **c–d** A dynamical network representation of the correlation of the GluN1b-GluN2B ATD interface residues bound to Fv2 (panel **c**) and Fv5 (panel **d**). The gray oval in panel **c** highlights the interaction between GluN1b ATD R1 and GluN2B ATD R2.

GluN2B ATDs (B/D subunits) are favored to interact with each other in Fv5-ATD but not in Fv2-ATD (Supplementary Fig. 9). This interaction likely stabilizes the open GluN2B ATD bi-lobe as in the nonactive2 conformation. Consistent with the cryo-EM structures, such changes in the ATDs alter the dynamics of inter-domain interactions between the ATD and LBD, and are ultimately relayed to the TMD to control channel activity.

Finally, we computed and analyzed an ensemble of optimal paths between the Fv2 binding site of the GluN2B ATD R1 and the GluN2B ATD R2 residues involved in stabilizing the closed bi-lobe in order to determine long-range dynamic networks that result in Fv2-mediated inhibition (Fig. 8a). Our analysis reveals that there are two major dynamic pathways that the binding of Fv2 strengthens: one from GluN2B ATD R1 to R2, and the other from GluN2B ATD R1 to GluN1b ATD R1 and into GluN2B ATD R2. These two pathways likely promote stabilization of the closed conformation of the GluN2B ATD. The specific GluN2B ATD R1-R2 pathways are represented by the GluN2B R1 regions corresponding to residues 131-136 (Fig. 8b), as well as residues 103-106 (Fig. 8c), GluN2B Tyr282 at the hinge of the bi-lobe (Fig. 8d), and the three GluN2B ATD R1-R2 "linkers": residues 146-149 (Fig. 8e), residues 283-288 (Fig. 8f), and residues 342-361 (Fig. 8g). The GluN2B ATD R1-GluN1b-ATD R1-GluN2B ATD

R2 pathway is generally characterized by interactions between GluN1b R1 residues 321, 337, 338, 340, 341, and 344 and GluN2B R2 residues 206-209 (Fig. 8h). These paths were also present in simulations of the Fv-free NMDAR in the ATD-closed conformation, indicating that the ATD conformation determines which routes are accessible for allosteric communication. In contrast to Fv2-ATD, the simulations of Fv5-ATD showed no optimal path between GluN2B ATD R1 residues 103-106 and R2. Additionally, there are significantly fewer paths between the GluN2B ATD R1, GluN1b-ATD R1, and GluN2B ATD R2 for Fv5-ATD than for Fv2-ATD (2/2585 unique paths for Fv5-ATD compared with 125/2331 unique paths for Fv2-ATD) (Fig. 8i). Overall, our computational analyses suggest that Fv2 binding to GluN2B ATD R1 facilitates closure of the GluN2B ATD bi-lobe by activating networks of long-range interactions, one within the GluN2B ATD and the other through inter-subunit interactions with the GluN1b ATD, thereby stabilizing the non-active1 conformation.

## Discussion

Here, we developed anti-GluN2B antibodies that can serve as lead molecular reagents to regulate the function of the NMDAR in a

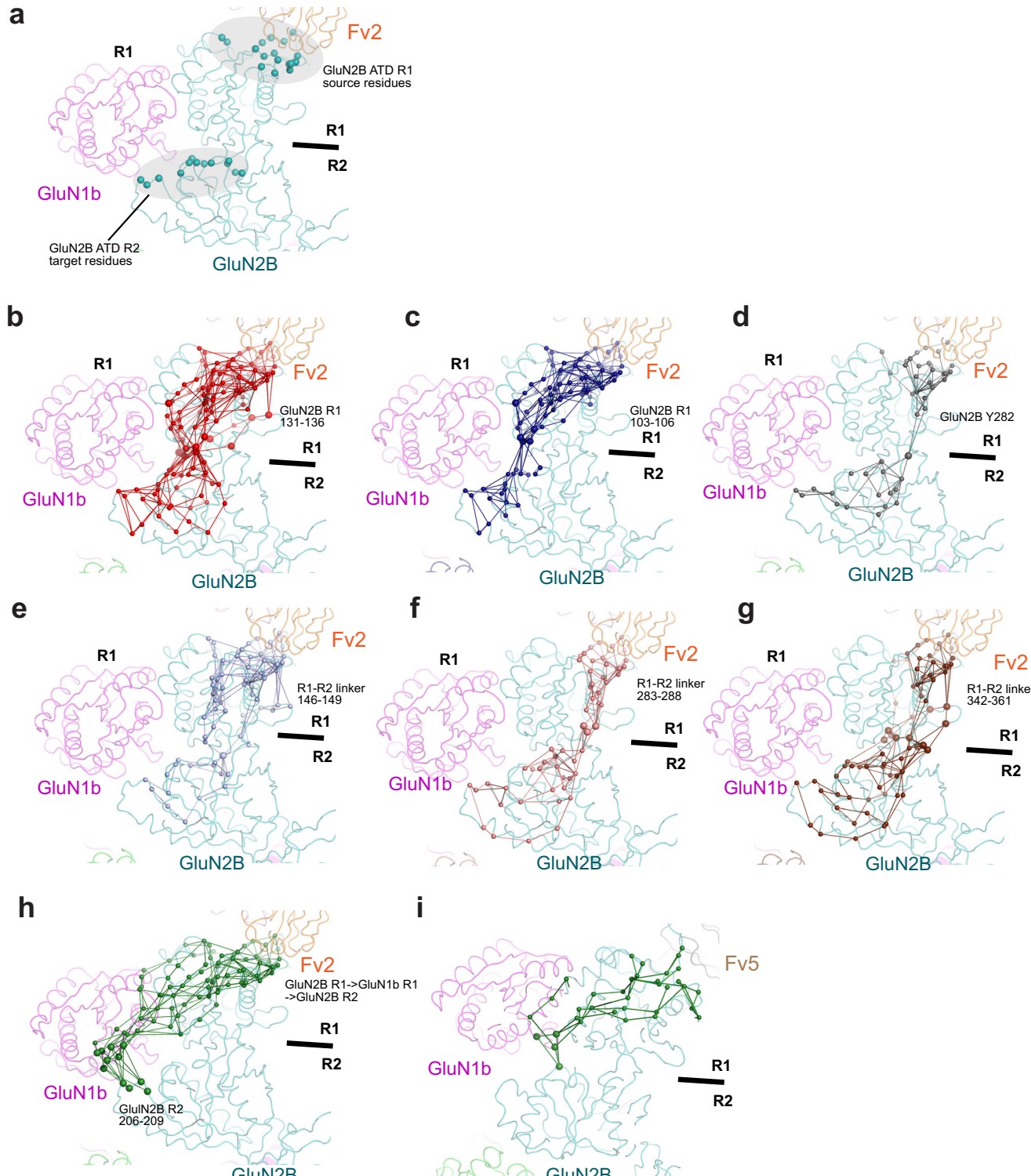

**Fig. 8 Fv2 binding and long-range interactions for bi-lobe closure of GluN2B ATD. a** GluN2B ATD R1 and R2 residues used as source and target nodes for computing optimal paths (surrounded by gray ovals). **b–d** Fv2 paths, involving GluN2B R1 residues 131-136 (panel **b**), GluN2B ATD R1 residues 103-106 (panel **c**) GluN2B Tyr282 at the R1–R2 hinge (panel **d**). **e–g** Fab2 paths transmitted through the R1-R2 linker residues 146-149 (panel **e**), 283-288 (panel **f**), 342-361 (panel **g**). **h** Fv2 paths that involve inter-subunit communication starting from GluN2B ATD R1 to GluN1b ATD R1, and into GluN2B ATD R2. **i** Same analysis for Fv5-ATD reveals one pathway that involves GluN1b-GluN2B contacts, although present in a very small percentage of the path ensemble. It is important to note that this path only involves a single point of contact on GluN1b, suggesting that correlated motion throughout GluN1b is specific to Fab2. All indicated residues are marked by large spheres.

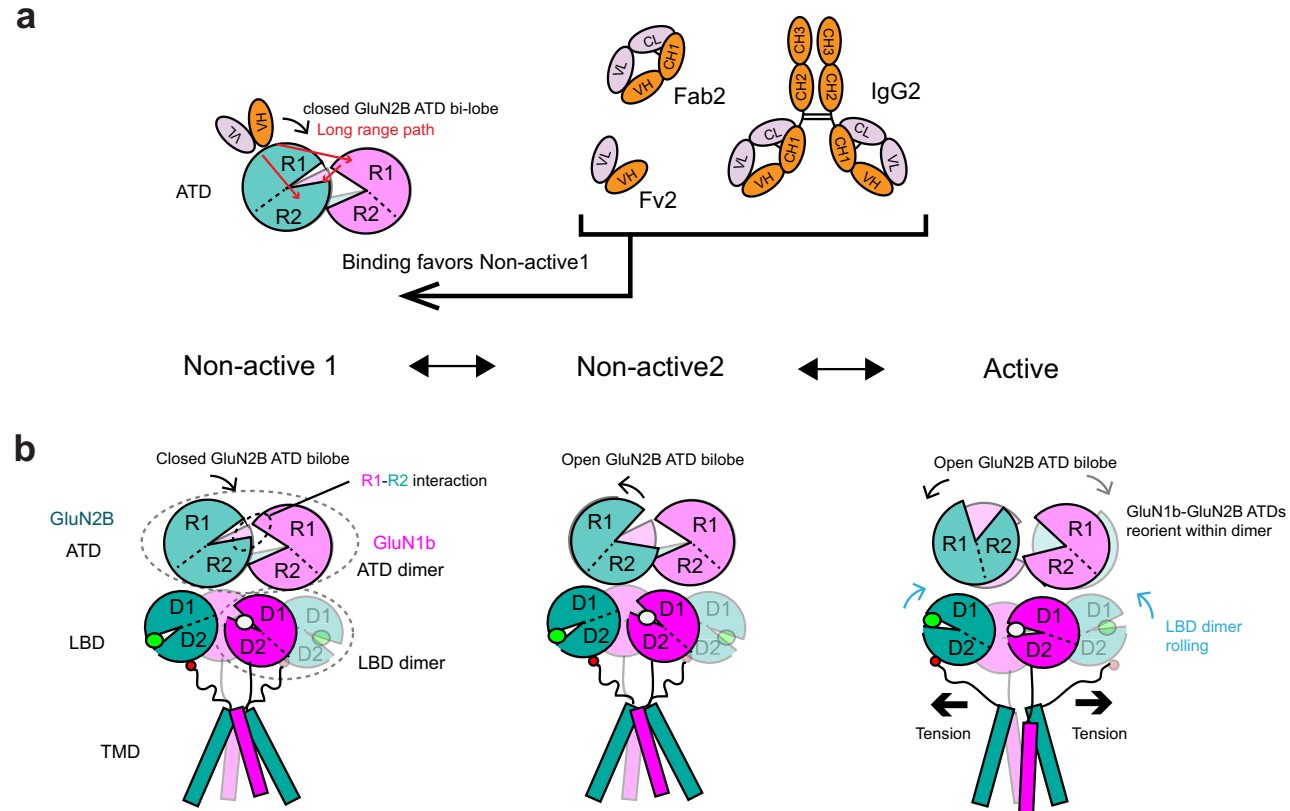

**Fig. 9 Potential mechanism of inhibition by Fv2, Fab2, or IgG2. a** Fv2, Fab2, and IgG2 bind to the R1 lobe of GluN2B ATD and stabilized the closed bi-lobe by two major long-range paths: one that directly goes from GluN2B ATD R1 to R2, and the other that goes from GluN2B ATD R1 to GluN1b ATD R1 and to GluN2B ATD R2 (red arrows). **b** The GluN2B ATD bi-lobe closure mediated by the long-range interactions shifts conformational equilibrium of the receptor toward non-active1, thereby favoring inhibition.

subtype-specific manner. One of the antibody clones, IgG2, recognizes the GluN2B ATD and allosterically inhibits the activity of the GluN1-GluN2B NMDAR channels by favoring closure of the GluN2B ATD bi-lobe as shown by our single-particle cryo-EM and x-ray crystallographic structures and all-atom molecular dynamics simulations of the extracellular structures complexed to Fv2. The stabilized conformation by IgG2 (or Fab2) mimics the one called non-active1[7,8], which is closely related to the conformation observed when bound to the negative allosteric modulator ifenprodil[22,24,29] (Fig. 9). While we cannot completely eliminate the possibility of receptor internalization, the inhibitory effect is likely elicited by conformational regulation of the NMDAR considering the time course of inhibition (less than 5 seconds) and the observation that the Fab and Fv fragments with no cross-linking ability shows clear inhibition. Whether Fv2 binds to the ATD-open conformation and rewires the contact map or instead selectively binds to the closed conformation and stabilizes that contact network has yet to be determined. Specific downregulation of GluN1-GluN2B NMDARs is highly desired since this subtype exists in the extrasynaptic space and mediates signaling for neurodegeneration[45]. Furthermore, GluN2B signaling is upregulated in the invasive front of a mouse model of pancreatic neuroendocrine tumorigenesis[3] and thus, downregulation of GluN1-GluN2B NMDAR may be useful in suppressing tumor invasiveness. The GluN2B specific inhibitory antibody developed here may be useful in studying neurodegeneration and tumor invasiveness and may serve as a therapeutic lead.

While IgG2 downregulates channel activity in a highly subtype-specific manner, it would be beneficial if one could

control potency and efficacy of inhibition. The cryo-EM structure of GluN1b-GluN2B NMDAR in complex with Fab2 provides some insights into how one may improve the inhibitory potency and efficacy by re-engineering IgG2, Fab2, or Fv2. For example, Arg101 in the H3 CDR loop could be mutated to a glutamate residue to promote a polar interaction with Arg67 of GluN2B. Furthermore, the L3 loop is in proximity to GluN2B, and the sequence could be altered. Last, the mini-loop between L1 and L2 may be extended and shuffled to promote binding with GluN2B. The efficacy and potency may also be improved by creating diabodies[46] in the context of either single chain Fab or Fv with the linker size adjusted to match the length of the two GluN2B subunits. Alternatively, diabodies may be made in combination with a single chain Fab or Fv against GluN1 ATD in order to stabilize the GluN1-GluN2B ATD heterodimer interface. This interface was shown by our simulation studies to be involved in one of the long-range dynamical interaction pathways that Fv2 mediates.

Recent studies showed that anti-NMDAR antibodies cause autoimmune diseases including anti-NMDAR encephalitis and Systemic lupus erythematosus (SLE). In anti-NMDAR encephalitis, anti-NMDAR antibodies have been shown to bind to the GluN1 subunit indicating that they target all subtypes of NMDARs. The sera containing anti-GluN1 antibodies was demonstrated to decrease synaptic currents[47]. In contrast, the SLE anti-NMDAR antibodies have been shown to bind either GluN2A or GluN2B but upregulate only the channels that contain GluN2A[48]. In both cases, antibodies target ATDs, but in a different region from the binding site of IgG2 and IgG5, which we identified in this work. The anti-NMDAR antibodies in

encephalitis and SLE have been shown to bind to the GluN1 ATD[49] and the hinge region of the GluN2A and GluN2B ATD bi-lobes[48], respectively. Thus, while the molecular mechanism is unclear, both of the natural autoimmune antibodies target ATDs. It is worth mentioning that the antibodies studied here were isolated from a mouse with no apparent behavioral changes, suggesting that anti-NMDAR immunizations do not necessarily elicit neuropsychiatric effects. Overall, our study shows that antibodies that bind to the ATD can allosterically inhibit NMDARs in a subtype-specific manner. These antibody-NMDAR structures inform re-engineering strategies for tuning the efficacy and potency of inhibition.

## Methods

**Preparation of anti-GluN1-GluN2B NMDAR IgGs.** Monoclonal antibodies (mouse immunoglobulin-γ (IgG)) that bind rat GluN1-GluN2B were obtained by immunizing mice with the purified intact GluN1a-GluN2B NMDAR proteins[29] using the standard protocol. We proceeded with hybridoma production using spleen from a mouse that did not show apparent alternation in neuropsychiatric behavior. Three other mice showed symptoms, most notably, tremor. Antigen binding was initially tested by ELISA on 96-well plates (MaxiSorp, Nunc) covered by the GluN1a-GluN2B NMDAR proteins in the presence of 0.005% LMNG. The positive clones were further tested by the Western blot analysis. IgGs were purified from hybridoma cell culture supernatants by rProtein-A Sepharose (GE Health-care). Fab fragments of the purified IgGs were obtained by papain proteolysis followed by re-running through rProtein-A Sepharose to remove the Fc fragment.

**Cloning and production of recombinant Fab5 and Fv2.** The heavy and light chains of Fab5 and Fv2 fragments were cloned from the hybridoma cells expressing IgG5 and IgG2, respectively. Total RNA was extracted using E.Z.N.A. Total RNA Kit I (OMEGA bio-tek). Reverse transcription was conducted using Easy First Strand Kit (Qiagen) and random hexamer primers. The VH and VL fragments of Fv2 or VH-C1 and VL-C fragments of Fab5 were amplified by primers with consensus sequences[50]. The heavy and light chains of Fv2 (Fv2H and Fv2L) were subcloned into the pNT-HisT vector (Takara Bio Inc.) and transformed into *Brevibacillus choshinensis* (Takara Bio Inc.) using the New Tris-PEG (NTP) method and plated onto MTNm agar (10 g Glycose, 10 g Polypeptone, 5 g Meat extract, 2 g Yeast extract, 10 mg FeSO$_4$ 7H$_2$O, 10 mg MnSO$_4$ 4H$_2$O, 1 mg ZnSO$_4$ 7H$_2$O, and MgCl$_2$ 6H$_2$O per liter). Colonies containing pNT-HisT-Fv2H and pNT-HisT-Fv2L were picked and co-cultured in MT liquid medium (15 g Agar, 10 g Glucose, 10 g Polypeptone, 5 g Meat Extract, 2 g Yeast extract, 10 mg FeSO$_4$ 7H$_2$O, 10 mg MnSO$_4$ 4H$_2$O, 1 mg ZnSO$_4$ 7H$_2$O, and MgCl$_2$ 6H$_2$O per liter) supplemented with 10 μg/ml Neomycin at 25º for 62 hours. The supernatant was concentrated and subjected to purification by Nickel-Chelating Sepharose (GE Healthcare) followed by Superdex200 (HiLoad16/600, GE Healthcare).

For recombinant expression of Fab5, the construct that starts with human alkaline phosphatase signal peptide, followed by the octa-Histidine tag, the light chain, the 57-amino acid long linker (57-link)[23], and the heavy chain was subcloned into the pFastBac vector (Thermo Fisher) for production of baculovirus by the Bac-to-Bac system (Thermo Fisher). High Five cells grown in ESF 921 media (Expression Systems) were infected with the recombinant baculovirus. The culture medium was collected 72 h post-infection, concentrated and purified using Cobalt-Chelating Sepharose (GE Healthcare) followed by trypsin digestion at 1:20 trypsin:Fab5 (w:w) ratio for 8 h at 18 °C to remove 57-link. The digested sample was further purified using Superdex200 (HiLoad16/600, GE Healthcare). Primer sequences are listed in a Supplementary Table 4.

**Fluorescence size exclusion chromatography (FSEC).** The purified IgG2 and 5, Fab2 and 5, or Fv2 were mixed and incubated with the purified GluN1b-GluN2A ATDs[23] or GluN1b-GluN2B ATDs[24] at a 3:1 weight ratio on ice for 1 h. The mixture was injected onto Superose 6 10/300 (GE healthcare) and the eluted proteins were detected by intrinsic Tryptophan fluorescence at the 280/320 nm excitation/emission wavelength. For structure-based analysis of IgG2 binding, GluN1a or GluN1b-EGFP fusion construct was coexpressed with GluN2B wildtype or mutant construct with the CTD deletion in HEK293 cells by transient trans-fection for 48 h. The cells were solubilized by 0.5% Lauryl Maltose Neopentyl Glycol (LMNG) for 1 h and ultracentrifuged at 98,000 g. The supernatant was mixed with IgG and subjected to FSEC using EGFP fluorescence (475/507 nm excitation/emission wavelength).

**X-ray crystallography of Fab2 and GluN1b-GluN2B-Fab5 complex.** The Fab2 proteins prepared by papain digestion of IgG2 as above were crystallized by vapor diffusion in hanging drop at 18 °C. Drops were prepared by mixing 1 μl of the purified Fab2 at 4 mg/ml with 1 μl of a solution containing 20% PEG3350 (w/w) and 8% Tascimate. Crystals were grown for seven days at 17 °C and frozen in the crystallization buffer supplemented with 20% glycerol. For GluN1b-GluN2B ATD

—Fab5 complex, 3:1 weight ratio of the recombinant Fab5 and the GluN1b-GluN2B ATD[22,24] was subjected to Superdex200 (HiLoad16/600, GE Healthcare). The fraction that contained the complex was concentrated to ~12 mg/ml, dialyzed against 10 mM Tris-HCl (pH 8), 50 mM NaCl, and 5 μM ifenprodil, and mixed with a half volume of reservoir solution (2–3 μl total drop size), which contained 2.1 M Na/K PO$_4$, 100 mM Li$_2$SO$_4$, and 100 mM CAPS (pH 10.5), and 4% for-mamide in a hanging-drop vapor diffusion configuration. The GluN1b-GluN2B-ATD Fab5 crystals were flash-frozen in liquid nitrogen in the presence of 20% glycerol. X-ray diffraction experiments were conducted at the wavelength of 1.0 Å at the 23ID-B beamline in the Advanced Photon Source in Argon National Laboratory for the GluN1a-GluN2B-ATD Fab5, and at the wavelength of 0.91 Å at the National Synchrotron Light Source II, Beam Line 17-ID-1 at Brookhaven National Laboratory. Diffraction data were processed using HKL2000[51]. The structures of Fab2 and GluN1a-GluN2B-ATD Fab5 were solved by molecular replacement using Fab from the PDB code 5B3J and GluN1b-GluN2B-ATD structure (PDB code: 3QEL) and the Fab from 5B3J, respectively, by using the program Phaser[52]. The model refinement was performed using the program Phenix[53].

**Cryo-EM of GluN1b-GluN2B-Fab2 and GluN1b-GluN2B-Fab5.** The GluN1b-GluN2B NMDAR proteins were expressed in *Spodoptera frugiperda* (Sf) 9 insect cells using the EarlyBac method[8,54]. The membrane fractions were solubilized by 0.5% LMNG at 4 °C for 1 h followed by ultracentrifugation at 98,000 g. The supernatant was subjected to purification by Strep-Tactin Sepharose (IBA) and the eluted fractions were mixed with the purified Fab2 or Fab5 at a 1:3 (w:w) ratio and incubated on ice for 1 h before further purification by Superose 6 10/300 (GE healthcare). The purified GluN1b-GluN2B NMDAR—Fab complexes at 3 mg/ml in 0.002% LMNG were mixed with 0.1% digitonin were placed on C-flat Holey Carbon Copper grids glow discharged for 25 s at 15 mA, and plunge-frozen in liquid ethane using a Vitrobot (FEI) at a relative humidity of 85% at 15 °C for 5 s blot time. Movies were collected on an FEI Titan-KRIOS microscope operating at 300 kV coupled with a post-GIF Gatan K2 Summit direct electron detector at 105k magnification (1.37 Å per pixel) with 70 total electrons in 50 frames over a 15 s exposure and with a defocus range of ~−1.5 μm and ~−3.0 μm. The movie pro-cessing, which included alignment, exposure weighting, and contrast transfer function estimation, was done by the program, WARP[55]. Single-particle analysis was conducted using the cisTEM workflow[56], which included 2D classification, ab-initio 3D reconstruction, and refinement by FrealignX[57]. To avoid over-refinement, the resolution limit for particle refinement was always set at least 2 Å lower than the resolution of the reconstructed structure as defined by the 0.143 cut-off of the Fourier shell correlation (FSC). The structural model of the GluN1b-GluN2B NMDAR (PDB ID: 6CNA[28]) and the crystal structure of Fab2 or Fab5 were docked into the cryo-EM density by the program Chimera[58] followed by remodeling by Rosetta[59]. The resulting model was refined against the cryo-EM map using Phenix real-space refinement[60]. Summary of data collection and refinement statistics are shown in Supplementary Table 1 and 3.

**Electrophysiology.** Recombinant GluN1-GluN2 NMDA receptors were expressed by co-injecting 0.05–1 ng of the wild-type or mutant rat GluN1-1b and GluN2 (A-D) cRNAs at a 1:2 ratio (w/w) into defolliculated *Xenopus laevis* oocytes. The current recording was performed by the two-electrode voltage-clamp recordings using agarose-tipped microelectrodes (0.4–1.0 MΩ) filled with 3 M KCl at a holding potential of -60 mV. During the measurements, the recording chamber was perfused with the bath solution containing 5 mM HEPES, 100 mM NaCl, 0.3 mM BaCl$_2$ and 10 mM Tricine at pH 7.4 (adjusted with KOH) at 18–22 °C. Currents were evoked by applications of 100 μM of glycine and L-glutamate. For patch-clamp experiments, HEK293T cells on glass coverslips were co-transfected with 700 ng pCI_neo GluN2B, 700 ng pIRE GluN1-1a or GluN1-1b, and 50 ng pEGFP and grew at 37 °C with 5% CO$_2$ in DMEM (Gibco) supplemented with 10% FBS. For expression of the rat triheteromeric GluN1-1a/GluN2A/GluN2B NMDAR, vectors kindly provided by Dr. Kasper Hansen[61], were transfected. Recordings were performed 24-h post-transfection using borosilicate glass capillaries (Sutter) pulled and polished to a final resistance of 2-6 MΩ when filled with the internal solution (composition in mM: 110 Cs-gluconate, 30 CsCl, 5 HEPES, 5 BAPTA, 4 NaCl, 2 MgCl$_2$, 0.5 CaCl$_2$, 2 ATP-Na, and 0.3 GTP-Na; pH 7.35). Cells were held at -80 mV and a rapid solution exchanger (RSC-200; Bio-Logic) was used to expose cells to an external buffer (composition in mM: 150 NaCl, 3 KCl, 10 HEPES, 1 CaCl$_2$, and 0.01 EDTA-Na (pH 7.4)) containing 100 μM each glycine and L-glutamate, with or without 0.1 mg/mL Fab or IgG. The bath chamber was extensively cleaned after each recording, and care was taken not to contaminate the bath with any Fab or IgG before initiating a recording. Data was collected on AxoPatch 200B patch-clamp amplifier (Axon Instruments), filtered at 2 kHz (Frequency Devices), and digitized with a Digidata 1550B digitizer (Axon Instruments) using a sampling frequency of 10 kHz. Recordings were analyzed using the Clampex 11.0 software (Axon Instruments). All experimental procedures related to *Xenopus laevis* were approved by the Institutional Animal Care and Use Committee of Cold Spring Harbor Laboratory (CSHL) and performed in accordance with the US National Institutes of Health (NIH) guidelines.

**Model construction and molecular dynamics simulations**. All-atom models of the Fab-bound and Fab-free NMDARs were constructed from the cryo-EM structures of Fab2 and Fab5 bound to the extracellular tetrameric NMDARs using MODELLER[62]. Since the constant regions of the Fab fragments were not well resolved in the cryo-EM structures, we used NMDAR-Fv models in this study. Agonists glutamate and glycine were modelled into the LBD binding sites using PDB ID 1PB7[16] for the glycine bound GluN1 LBD and PDB ID 2A5S[14] for the glutamate bound GluN2 LBD, since they were not resolved in the cryo-EM structures. In order to satisfy the system size limit of 700,000 atoms on the special-purpose supercomputer Anton 2[63], the ATD-LBD-TMD cryo-EM constructs were truncated to include only the extracellular domains (ATD and LBD). A Gly-Thr dipeptide linker was modeled to bridge the S1 and S2 segments in lieu of the LBD-TMD linkers present in the intact receptor[14]. Systems were solvated and ions were added to neutralize the systems and bring them to 150 mM NaCl using CHARMM-GUI[64]. An octahedral solvent box of dimensions 248 Å × 172 Å × 172 Å (691,605 total atoms) for Fv2 and 234 Å × 162 Å × 162 Å (578,971 total atoms) for Fab5, and 180 Å × 180 Å × 180 Å for the Fv-free NMDAR were used. Production equilibrium molecular dynamics simulations were performed using Anton 2. Prior to simulation on Anton 2, equilibration and pre-production simulations were performed using NAMD 2.13[65]. Equilibration was performed first at NVT with a gradually relaxing set of backbone-sidechain restraints and then at NPT without restraints. Pre-production simulations were continued at 310 K with a constant pressure of 1 atm and a timestep of 2 fs. Output (coordinates, velocities, and extended system information) from preproduction simulations was extracted after 5 ns at 0.25 ns intervals and used as starting states for a series of twenty 50 ns replicas (ten for Fv2 and ten for Fv5) simulated on Anton 2 totaling one microsecond of simulation time. The Fv-free and Fv2 system with the alternative ATD-LBD linker conformation were each simulated for five 50 ns replicas. On Anton 2, all replicas were run with the default timestep of 2.5 fs at 310 K. Trajectories were unwrapped/re-wrapped using the PBCTools plugin of VMD[66]. To remove translational and rotational protein motion, all replica frames were aligned by the backbone atoms of the NMDAR (no antibodies) to a reference structure using the MDAnalysis python library[67,68]. Each replica was split into two 25 ns windows for analysis for a total of 20 simulation windows for each Fv (10 simulation windows for the Fv-free system).

**Analysis of molecular dynamics simulation trajectories**. Molecular dynamics trajectories were used to construct a dynamical network representation of each Fv-bound NMDAR complex and the Fv-free complex in the closed-ATD conformation using the dynetan python implementation of dynamical network analysis[42]. Network topology was generated using one node for every Cα atom, drawing an edge between each pair of nodes where the heavy atoms of both residues are within 6 Å of each other for ≥75% of the trajectory. Since this work focuses on interface interactions, 6 Å was selected instead of the default 4.5 Å to ensure that all relevant interface interactions are captured. To quantify the strength of each edge pair, the generalized correlation coefficient[42] was computed using the equation below. These generalized correlation coefficients are obtained by calculating a mutual information estimator $I$ to quantify the information shared between two residues Cα's through atomic fluctuations and using $I$ to compute the generalized correlation coefficient $r_{MI}$ using the following equation for $d = 3$ dimensions:

$$r_{MI} = \left(1 - e^{-2I/d}\right)^{1/2} \quad (1)$$

Using a generalized correlation is advantageous because it incorporates non-linear relationships between node fluctuations and accounts for correlations between nodes fluctuating together along perpendicular axes of motion[69].

While one advantage of performing short 50 ns replica simulations is the ability to capture diverse local conformational dynamics of the cryo-EM structures[70], a potential limitation of this approach is that some key structural rearrangements that result in stable interface contacts may not form in time to meet the 75% contact frequency criterion for every window. When computing the average correlation coefficient across windows, including windows in which the interaction is absent (zero correlation) might underrepresent the degree of correlation that occurs when the interaction is present. In addition to computing correlation coefficients and their standard errors for all windows, correlation coefficients and their standard errors were also computed for windows with nonzero correlation, and the number of windows in which correlated motion was detected was reported.

In addition to studying interface interactions, the correlation coefficients of the dynamical network were used to determine the optimal paths between the Fv binding site and the GluN2B R2 lobe. The Floyd–Warshall algorithm as implemented in the dynetan[42] and NetworkX[71] libraries was used to compute optimal paths between sets of source and target nodes for each window. Source nodes were selected as residues on the GluN2B ATD R1 lobe with which our simulations reveal Fv2 is most highly coupled ($r_{MI} > 0.3$ for ≥15 windows), and target nodes were selected as residues on the GluN2B R2 lobe that contact the GluN2B ATD R1 lobe or the GluN1b ATD R1 lobe as revealed by the dynamical network. This results in an ensemble of unique optimal paths between all possible combinations of source and target nodes. These paths were then sorted by common edges between GluN2B R2 and either GluN2B R1 or GluN1b R1 for analysis. To reduce the amount of noise in the optimal path dataset for each Fv, a filter was applied to the optimal paths for each pair of source and target residues that only

considered paths present in more than one simulation window in addition to all unique paths exhibiting a Jaccard similarity coefficient that exceeds a threshold defined as the maximum Jaccard similarity coefficient for which all paths have at least one other similar path in the path ensemble (≥0.6 for Fv2 and Fv5 and ≥0.4 for the Fv-free system).

To quantify the range of binding modes observed for the Fv fragments, simulation trajectories were clustered using the mean-shift algorithm[43] as implemented in Scikit learn[72]. The mean-shift algorithm uses the density of points to define cluster centers in feature space. This algorithm is suitable for conformational clustering, as it does not require prior knowledge or estimation of the number of expected clusters. Here, a two-dimensional feature set was chosen as the Cα RMSD of the heavy (dimension one) and light (dimension two) chain CDR loops of each Fv. Prior to clustering, all replica frames were aligned by the Cα atoms of the top lobe of GluN2B (chain B, residues 34-145 and 289-341). The mean shift bandwidth was selected to be 2.5 for both Fv fragments, which produced clusters with spatially distinct secondary structure orientation and CDR loop positions.

The contact surface area between the Fvs and the ATDs was determined using the difference in solvent accessible surface area $S$ between the isolated ATDs and Fvs and the Fv-bound ATD complex[44].

$$(S_{ATDs} + S_{Fabs} - S_{complex})/2 \quad (2)$$

The Shrake-Rupley algorithm[73] for the solvent-accessible surface area as implemented in the MDTraj python package[74] was used for computing the terms of the contact surface area with a probe radius of 0.17 nm, the van der Waals radius of water[75].

**Reporting summary**. Further information on research design is available in the Nature Research Reporting Summary linked to this article.

## Data availability

Cryo-EM density maps and atomic coordinates for NMDAR-Fab2, NMDAR-Fab5 have been deposited in the electron microscopy data bank and the Protein Data Bank. For NMDAR-Fab2 the accession codes are EMD-25843 and PDB 7TE9 (non-active1); EMD-25844 and PDB 7TEB (non-active1-like); EMD-25845 and PDB 7TEE (non-active2-like). For NMDAR-Fab5 they are EMD-25849 and PDB 7TEQ (active); EMD-25850 and PDB 7TER (non-active2); EMD-25851 and PDB 7TES (non-active1); EMD-25852 and PDB 7TET (non-active2-like). X-ray crystallographic data and coordinates of Fab2 and GluN1b-2B ATD-Fab5 have been deposited to the Protein Data Bank (PDB) under accession codes 7TE4 and 7TE6, respectively. Data points for electrophysiology are available as Source data. Source data are provided with this paper.

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

## Acknowledgements

We thank the staff at the 23-ID beamlines at the Advanced Photon Source at the Argonne National Laboratory for support during data collection. Dennis Thomas and Ming Wang are thanked for managing the cryo-EM facility and the computing facility, respectively. Carmelita Bautista is thanked for the excellent facility service for producing hybridoma cell-lines. We thank Dr. Kasper Hansen for providing expression vectors for triheteromeric GluN1-2A-2B NMDAR. Anton 2 computer time (MCB130045P) was provided by the Pittsburgh Supercomputing Center (PSC) through NIH grant R01GM116961 (to A.Y.L.); the Anton 2 machine at PSC was generously made available by D.E. Shaw Research. We also used resources provided by the Maryland Advanced Research Computing Center (MARCC) at Johns Hopkins University. This work was funded by the NIH (NS111745 and MH085926 to H.F.), Robertson funds at Cold Spring Harbor Laboratory, Doug Fox Alzheimer's fund, Austin's purpose, Heartfelt Wing Alzheimer's fund, and the Gertrude and Louis Feil Family Trust (all to H.F.); NIH F32MH121061 (to K.M.) and T32GM135131 (to R.A.Y.); and Johns Hopkins Catalyst Award (to A.Y.L.).

## Author contributions

N.S. and H.F. designed and conducted experiments related to antibody screening. N.T. conducted experiments involving cryo-EM and the Fab2 crystallography. M.R. solved the structure of GluN1b-GluN2B ATD-Fab5 by x-ray crystallography. N.T. conducted two-electrode voltage clamp electrophysiology. K.M. and R.G. conducted patch clamp electrophysiology. R.A.Y. conducted molecular dynamics simulations and dynamical network modeling; R.A.Y. and A.Y.L. analyzed the results. N.T., R.A.Y., A.Y.L., and H.F. wrote the manuscript.

## Competing interests

The authors declare no competing interests.
