## [Peer Review File · Nature Communications]

Development and characterization of functional antibodies targeting NMDA receptorsREVIEWER COMMENTS

Reviewer #1 (Remarks to the Author):

The use of targeted antibodies to modulate the activity of proteins either in the clinic or as a tool to investigate their normal function is extremely promising. In the present study, the authors develop antibodies against the NMDA receptors, specifically those containing the GluN2B subunit. They identify and isolate one monoclonal Ab, IgG2, that inhibits the activity of GluN1-GluN2B NMDA receptors, and another, IgG5, that weakly enhances their activity. This manuscript is highly significant because it identifies a novel means to specifically target NMDA receptors.

The authors use a variety of techniques including electrophysiology, size-exclusion chromatography, single particle cryo-EM, molecular dynamics simulations, mutagenesis, etc. to verify the action of these antibodies and to demonstrate that a direct interaction between the Abs and the ATD of the GluN2B subunit mediate their action. The approaches and the experiments are rigorous and provide solid and clear conclusions. The authors are cautious in not over interpreting their data.

Overall, the manuscript is extremely strong and rigorous and will be of interest to a broad audience. Not only those interested in NMDA receptors but the general question of modulation of protein function by antibodies, given the exploration of the basis of how IgG2 and IgG5 alter receptor function. I only have some largely minor questions and concerns.

Specific comments.

1. In the brain, NMDA receptors are typically triheteromeric receptors and the effect of subunit-specific modulators is often difficult to predict without directly testing. Although the authors have done a heroic job in terms of addressing the mechanism of action of IgG2 and IgG5, there would be significant advantages in terms of their in vivo actions as well as their mechanism of action to address the effect of at least IgG2 on GluN1-GluN2A-GluN2B receptors.

2. Out of curiosity and not asking that they be added to manuscript but were any of the 'unfolded' Ab tested for their effects on receptor function?

Minor comments.

1. Figure 4C. Is not IgG2 inducing a weak enhancement of activity in GluN1-4b-GluN2A? Also in Figure 4A, purification is misspelled in cartoon.

2. Page 6. 'Also, the Fv2 fragment...'. I guess you are comparing 60.0 +/- 3.2% to 54.2 +/- 5.5%? Is this statistically significant? Also, not clear what is being stated in this section of writing.

3. Fab5 has extensive interactions with the GluN1 ATD and to a lesser extent GluN2B ATD (Figure 4B & MD simulations), yet the binding site for Fab5 on GluN2B ATD is emphasized (Figure 4D and 4F). Is there a reason the sites for GluN1 ATD are not discussed further in the structure?

4. The changes in distribution of occupancy of the different non-active states and active states, with Chou et al., 2020 as a reference, is very interesting (Figures 5 and 6). Would a table summarizing these distributions in the various states be useful? Or misleading.

5. Figure 7. 'The specific value of the correlation coefficient between...'. Cannot really see any difference between the lines?

Reviewer #2 (Remarks to the Author):

Here Tajima et al. present a discovery and characterization study on NMDA receptor antibodies. They generate mouse monoclonal IgGs by immunizing mice with purified NMDA receptors and identify two high affinity mAbs selective for the GluN2B subunit. They do a thorough workup on subunit selectivity and activity and find that "IgG2" is a moderately efficacious subtype selective receptor antagonist, by both TEVC and patch clamping HEK cells. They then carry out crystallographic and cryo-EM based structural studies to define the details of the interactions between the Fab version of the IgG and the receptor; these results, while not at very high resolution for the cryo-EM, are convincing, and are well supported by mutagenesis and electrophysiology. The EM structures are further used to understand the conformational states stabilized by the antibody fragments. MD simulations are used to flesh out the mechanisms by which the Fab2 / IgG2 antagonizes the receptor. Overall the manuscript is clearly written and presented and the topic will be of broad interest to communities working on iGluRs and on developing antibody tools for either native receptor purification or for therapeutic leads. I have no major concerns, only minor comments, some of which are stylistic, and the authors can choose to ignore.

Specific comments/questions:

The repeated intro statement on lines 46 and 52, that these domains “mediate functions,” would be more impactful if made more specific.

Lines 60-64. I am generally curious why most of these drugs targeting NMDA receptors have not worked out. A couple references here would be helpful.

Line 75 and 205: a (not the) non-active conformation. Can these non-active conformations be related to resting-like or desensitized-like conformations? Relevant to lines 316-317 also. If they are entirely distinct from resting or desensitized states, that is interesting- suggests either an induced fit kind of mechanism for binding, rather than conformational selection. If the non active conformations can be related to known functional states, it would help in developing an intuitive understanding of how they antagonize channel activity.

Line 176, amenable (not prone) to mutagenesis.

Did the mice used to generate the antibodies exhibit any obvious signs of encephalitis / convulsant disorders? Could be mentioned in the Methods. I bring it up because lack of animal survival presents a challenge for people trying to make antibody tools against major neurotransmitter receptors; if you found a way to overcome this, it could be broadly impactful.

Figure 1D: is the current decay just due to rundown?

Fig. 4 implicates that Fab2 and 5 bind to non-overlapping sites. Do they bind competitively?

Reviewer #3 (Remarks to the Author):

Tajima and colleagues have taken a multi-disciplinary approach to identify and then study the mechanism by which an antibody (i.e. IgG2) selectively and functionally targets GluN2B-containing NMDA receptor subunits. Using electrophysiology experiments, the authors show that the antibody downregulates channel activity by binding to the amino-terminal domain of the GluN2B receptor subunit and, in doing so, promotes a non-active conformational state of the channel.

The work from Furukawa and Lau labs is expertly performed with the biochemical/structural data and molecular dynamic simulations providing interesting and detailed information about the antibody-channel interactions. Despite these positive notes, as outlined below the study however lacks novelty since a number of other high profile studies have established that antibodies can target NMDA receptor in a subunit-dependent manner. Also, the functional mechanism by which antibody binding promotes channel closure needs to be studied in more detail as well as some basic pharmacology questions which are outlined in more detail below. In keeping with this, the authors need to make a better case that different conformational states of the NMDA receptor can be actually linked to an inactive/closed state of the channel.

In summary, although the structural and computational work in the manuscript provides a valuable framework in which to understand how NMDA receptors may be targeted by antibodies, such as in lupus, as it currently stands, the manuscript is better suited to a more specialist journal.

Major:

1. The authors' main take-home message, as stated in the Abstract, is that their observations demonstrate a proof of concept that antibodies may serve as specific reagents to regulate NMDA receptor function for basic research and therapeutic objectives. The veracity of that claim is, however, challenged by a number of previous studies that have already shown, for example, that autoantibodies can target both GluN1 and GluN2 subunit of NMDA receptors (see Wollmuth et al, 2021; Chan et al, 2020, Jones et al, 2019 amongst others). So, the author's main claim is not that new and nor do they link their findings into a neurological disease or specific therapeutic benefit. The Chan et al study nicely demonstrates that autoantibodies to the NMDA receptor GluN2A subunit potentiates the response and leads to some of the negative neurological features reported in patients with lupus. Given all of this, it is not clear to me what key finding is new for the broader readership of the Journal.

If the authors really wish to pursue this rationale to justify their study, I think they have to overcome some other important issues. In terms of basic science questions, genetic approaches (knockout, knockin and conditional mice) have been extremely powerful techniques in studying NMDA receptors (and other proteins for that matter) so it is not clear how antibodies would offer a superior and/or useful alternative approach. In terms of the clinic, as noted above, autoantibodies targeting NMDA receptors are rather associated with negative neurological features rather than being helpful therapeutics. I appreciate that there is significant excitement around antibodies in the treatment of Alzheimer's disease (as mentioned in the Introduction) but the efficacy of targeting Abeta with antibodies seems to be quite small and some clinicians/researchers have even questioned the validity of the findings. Neurological disease that is linked to NMDA receptor dysfunction isn't usually just a matter of downregulating receptor function as purported to be the case in Alzheimer's. And even if that

approach was possible, different NMDA receptor subunits are not neatly expressed in distinct neuronal populations or neuronal circuits so the issue of specificity would still remain unresolved. Benzodiazepines have proven to be useful in targeting neurological function (e.g. anxiety, muscle relaxation, epilepsy etc) because GABA receptor alpha subunits are discretely expressed in the CNS and are inadvertently tied into distinct behaviors and disease (see papers from Rudolph/Mohler and/or Fritschy labs). So far, this does not seem to be the case for NMDA receptors or other iGluRs for that matter. Consequently, it is not immediately clear how the authors' work offers a significant advance.

2. The authors need to make a more careful pharmacological analysis of their findings. For example, is inhibition of the NMDA receptor by antibodies reversible or, as most would suspect, irreversible. In the latter case, how would irreversible "block" of the receptor be advantageous since most pharmacologists tend to think of long-lasting or irreversible block to be problematic. After inhibition, is the receptor endocytosed by the cell/neuron. Also, the authors need to show that the antibody has an effect on native channels, ideally by using acutely isolated brain slice tissue. Synaptic activation of NMDA receptors requires L-Glu to be present in the synaptic cleft for only 1 ms or so. Is antibody binding able to exert inhibition under these more rapid experimental conditions? Also, does the antibody cross the blood brain barrier?

3. Using distinct structural states identified in previous studies from the Furukawa lab (Tajima et al, (2016) Nature and Chou et al (2020) Cell), the authors conclude that IgG2 antibody binding promotes preferential occupancy of a specific non-active state (i.e. non-active1) of the NMDA receptor. The authors note further that this state most closely corresponds to the state adopted by the NMDA receptor when bound to the negative allosteric modulator, ifenprodil. The authors however do not attempt to link these two findings. For example, does pre-incubation with ifenprodil oppose the ability of IgG2 from inhibiting GluN2B-containing NMDA receptors or vice versa? Also, how do the different inactive state of the channel, proposed in this study and in previous studies, relate to the closed states measured by single channel analysis (Yuan et al, 2009 amongst others). Given the broad readership of the Journal, it would be important to establish or at least propose a link between the structural information with functional characterization of the receptor.

Minor:

1. The authors identify the different NMDA receptor subunits GluN1, GluN2A-D and GluN3A-B, however, they fail to define the splice variants, GluN1a and GluN1b. Some clarification is needed.

2. There is no clarification in the main text for the abbreviations, Fv and Fab, which should be addressed.

Reviewer #5 (Remarks to the Author):

Tajima et al. combined functional and structural studies with molecular dynamics simulations to show that an antibody binds to a non-active conformation of the amino-terminal domain of the GluN1/GluN2B subtype of NMDA receptors to allosterically down-regulate the channel activity, but has no effects on other subtypes (N2A, N2C, and N2D). This work is important in demonstrating antibodies as potential therapeutics for subtype-specific targeting. I have the following comments for improving the paper.

1. The functional, structural, and molecular simulation results generally support and reinforce each other rather well. One place where the degree of complementarity falls short is the binding-site mutations. Here they made a speculation: "When the three residues were mutated to alanine, inhibition of IgG2 was mostly but not completely removed (Fig. 3E), perhaps indicating that the alanine mutations did not completely mask the binding capability..." Instead of speculation, the authors could carry out a binding assay of the mutants (as they did for wild-type in Fig. 2).

2. Regarding Ig5, the authors made contradictory claims. In p. 5, the minor potentiating effect of IgG5 was touted as an example of upregulation. However, in p. 11, when explaining their cryo-EM data, they stated the data were consistent with little or no functional effects of IgG5.

3. p. 10, "Although both IgG2 and IgG5 bind the GluN2B ATD, they 203 do so at different protein surfaces (Fig. 4E-F)." It will be really helpful if the authors elaborate on these different binding surfaces.

Minor: p. 9, "Fig. 3E-H" -- should H be F?

p. 12, "First, application ... show" -- subject-verb mismatch.

REVIEWER COMMENTS

Reviewer #1 (Remarks to the Author):

1. In the brain, NMDA receptors are typically triheteromeric receptors and the effect of subunit-specific modulators is often difficult to predict without directly testing. Although the authors have done a heroic job in terms of addressing the mechanism of action of IgG2 and IgG5, there would be significant advantages in terms of their in vivo actions as well as their mechanism of action to address the effect of at least IgG2 on GluN1-GluN2A-GluN2B receptors.

We agree with the reviewers on the importance of the tri-heteromeric GluN1-2A-2B NMDARs. Thus, we conducted the experiments on the tri-heteromeric receptors using the method developed by Hansen et al. using the patch-clamp method. Indeed, we see a milder inhibitory effect compared to the di-heteromeric GluN1-2B NMDARs, indicating that the number of binding sites matters for controlling the extent of inhibition. This result is included in the revised manuscript (Fig. S2).

2. Out of curiosity and not asking that they be added to manuscript but were any of the 'unfolded' Ab tested for their effects on receptor function?

That sounds interesting. No, we have not been able to test that. What we meant by 'folding specific' is folding of NMDA receptors, not IgG. When we unfold NMDA receptors with SDS in Western blotting, we do not detect NMDA receptors with IgG2 or IgG5. We clarified this more clearly in the revised text (pg 5).

Minor comments.

1. Figure 4C. Is not IgG2 inducing a weak enhancement of activity in GluN1-4b-GluN2A? Also in Figure 4A, purification is misspelled in cartoon.

That is a great observation. It is slightly but not statistically significant. We mentioned this point in the revised manuscript. Also, the misspelling was corrected. Thank you.

2. Page 6. 'Also, the Fv2 fragment...'. I guess you are comparing 60.0 +/- 3.2% to 54.2 +/- 5.5%? Is this statistically significant? Also, not clear what is being stated in this section of writing.

We thank the reviewers to pointing this out. They are indeed, insignificant. The text on pg6 is rewritten accordingly.

3. Fab5 has extensive interactions with the GluN1 ATD and to a lesser extent GluN2B ATD (Figure 4B & MD simulations), yet the binding site for Fab5 on GluN2B ATD is emphasized

(Figure 4D and 4F). Is there a reason the sites for GluN1 ATD are not discussed further in the structure?

Fab5 actually has extensive interactions with GluN2B ATD and to a lesser extent with GluN1. We added minor interface on GluN1b in the revised Fig. 4F.

4. The changes in distribution of occupancy of the different non-active states and active states, with Chou et al., 2020 as a reference, is very interesting (Figures 5 and 6). Would a table summarizing these distributions in the various states be useful? Or misleading.

Thank you for your suggestion. We made a Table (please see below). This information is already embedded in Fig. 5 and 6. We will incorporate it if the editors feel it's appropriate.

	Non-active 1 (%)	Non-active 2 (%)	Active (%)
Fab2	82.9	17.1	0
Fab5	29.5	63.1	7.4

5. Figure 7. 'The specific value of the correlation coefficient between...'. Cannot really see any difference between the lines?

Mostly there are not many changes. However, you probably see different line colors (e.g., red, black, grey) especially in panel c and d in Fig. 7. We changed the patterns of transparency and fog in Fig. 7 to improve the color contrast.

Reviewer #2 (Remarks to the Author):

The repeated intro statement on lines 46 and 52, that these domains "mediate functions," would be more impactful if made more specific.

We agree. We revised the manuscript in the following ways: "...manners to mediate functions including channel gating, allosteric modulation, and cellular signaling" (line 46) and "...discrete patterns to control channel gating and allosteric modulation"

Lines 60-64. I am generally curious why most of these drugs targeting NMDA receptors have not worked out. A couple references here would be helpful.

There are some effective drugs that target NMDA receptors including memantine and ketamine. Having said that, there are many compounds that bind to NMDA receptors that failed in clinical trials. We added appropriate citations to address this (Lipton and Lechslaw).

Line 75 and 205: a (not the) non-active conformation. Can these non-active conformations be related to resting-like or desensitized-like conformations? Relevant to lines 316-317 also. If they are entirely distinct from resting or desensitized states, that is interesting- suggests either an

induced fit kind of mechanism for binding, rather than conformational selection. If the non active conformations can be related to known functional states, it would help in developing an intuitive understanding of how they antagonize channel activity.

That is a very good point. Indeed, we have not been able to definitively distinguish between closed channel vs. desensitized state.

Line 176, amenable (not prone) to mutagenesis.

We agree. Revised accordingly.

Did the mice used to generate the antibodies exhibit any obvious signs of encephalitis / convulsant disorders? Could be mentioned in the Methods. I bring it up because lack of animal survival presents a challenge for people trying to make antibody tools against major neurotransmitter receptors; if you found a way to overcome this, it could be broadly impactful.

This is a great question. We selected animals that survived the injection of the GluN1-2B NMDA receptor proteins. While we did not conduct extensive behavioral assay on the mice, we did not notice anything drastic changes for this animal. There are three other mice that did not survive. We mentioned this point in the method section of the revised manuscript.

Figure 1D: is the current decay just due to rundown?

Yes. We have slight rundowns in GluN1/2C and GluN1/2D. Not so much for GluN1/2A and GluN1/2B in our system.

Fig. 4 implicates that Fab2 and 5 bind to non-overlapping sites. Do they bind competitively?

No they do not likely bind competitively since the binding sites do not overlap as you mentioned. Indeed, we can purify GluN1b-2B NMDAR in complex with both Fab2 and Fab5.

Reviewer #3 (Remarks to the Author):

1. The authors' main take-home message, as stated in the Abstract, is that their observations demonstrate a proof of concept that antibodies may serve as specific reagents to regulate NMDA receptor function for basic research and therapeutic objectives. The veracity of that claim is, however, challenged by a number of previous studies that have already shown, for example, that autoantibodies can target both GluN1 and GluN2 subunit of NMDA receptors (see Wollmuth et al, 2021; Chan et al, 2020, Jones et al, 2019 amongst others). So, the author's main claim is not that new and nor do they link their findings into a neurological disease or specific therapeutic benefit. The Chan et al study nicely demonstrates that autoantibodies to the NMDA receptor GluN2A subunit potentiates the response and leads to some of the negative neurological features reported in patients with lupus. Given all of this, it is not clear to me what key finding is new for the broader readership of the Journal.

Indeed, the articles by Wollmuth/Diamond, and Jones et al are of high quality and represent significant advances in the autoimmunity field. However, our work here does not have much to do with autoimmune antibodies, instead, our aim here is to isolate a lead reagent that we can build around for eventual development of effective subtype-selective reagent. Thus, the current work has a different scope and does not compete against the previous work that the reviewer listed. We removed 'proof-of-concept' from the abstract.

If the authors really wish to pursue this rationale to justify their study, I think they have to overcome some other important issues. In terms of basic science questions, genetic approaches (knockout, knockin and conditional mice) have been extremely powerful techniques in studying NMDA receptors (and other proteins for that matter) so it is not clear how antibodies would offer a superior and/or useful alternative approach. In terms of the clinic, as noted above, autoantibodies targeting NMDA receptors are rather associated with negative neurological features rather than being helpful therapeutics. I appreciate that there is significant excitement around antibodies in the treatment of Alzheimer's disease (as mentioned in the Introduction) but the efficacy of targeting Abeta with antibodies seems to be quite small and some clinicians/researchers have even questioned the validity of the findings. Neurological disease that is linked to NMDA receptor dysfunction isn't usually just a matter of downregulating receptor function as purported to be the case in Alzheimer's. And even if that approach was possible, different NMDA receptor subunits are not neatly expressed in distinct neuronal populations or neuronal circuits so the issue of specificity would still remain unresolved. Benzodiazepines have proven to be useful in targeting neurological function (e.g. anxiety, muscle relaxation, epilepsy etc) because GABA receptor alpha subunits are discretely expressed in the CNS and are inadvertently tied into distinct behaviors and disease (see papers from Rudolph/Mohler and/or Fritschy labs). So far, this does not seem to be the case for NMDA receptors or other iGluRs for that matter. Consequently, it is not immediately clear how the authors' work offers a significant advance.

We thank the reviewer for the comments. We appreciate the reviewer pointing out that the rationale for targeting NMDA receptor is weaker compared to the extremely successful case of benzodiazepine on GABA receptor. We also understand that the effect of Abeta targeting antibodies on Alzheimer's disease is not clear as of 2021.

Having said that, pharmacological targeting of NMDA receptors has shown promises in treating neurological conditions including the ones for ketamine, memantine, and possibly glyx13. Other pharmacological reagents against NMDA receptors, although not therapeutically effective, prove to be important in basic research. Thus, we believe that finding different means to target NMDAR is a justifiable investment to make.

Lastly, we wanted to mention that these antibodies were isolated from an animal that survived the NMDA receptor immunization. Although we could not conduct a thorough behavioral assay, we have not noticed abnormality in its behavior.

2. The authors need to make a more careful pharmacological analysis of their findings. For example, is inhibition of the NMDA receptor by antibodies reversible or, as most would suspect, irreversible. In the latter case, how would irreversible "block" of the receptor be advantageous

since most pharmacologists tend to think of long-lasting or irreversible block to be problematic. After inhibition, is the receptor endocytosed by the cell/neuron. Also, the authors need to show that the antibody has an effect on native channels, ideally by using acutely isolated brain slice tissue. Synaptic activation of NMDA receptors requires L-Glu to be present in the synaptic cleft for only 1 ms or so. Is antibody binding able to exert inhibition under these more rapid experimental conditions? Also, does the antibody cross the blood brain barrier?

Thanks for the suggestions. We do not think that the majority of the effect is caused by endocytosis since Fab and Fv fragments can induce inhibition in a short time scale in both oocytes and HEK293 cells. As a matter of fact, we have not been able to observe clear sign of endocytosis using confocal microscopy at this point.

We are looking into blood-brain barrier crossing and this one does not appear to cross (also preliminary at this point). The inhibitory effect is long lasting but can be reversed partially (it required longer time than sustainable time period of electrophysiology experiments).

Indeed, we are working toward structure-based engineering IgG2 to have faster off-rate. Results from the above lines of research are beyond the scope of the current manuscript that already contains the antibody discovery, characterization, structures, and MD simulations.

3. Using distinct structural states identified in previous studies from the Furukawa lab (Tajima et al, (2016) Nature and Chou et al (2020) Cell), the authors conclude that IgG2 antibody binding promotes preferential occupancy of a specific non-active state (i.e. non-active1) of the NMDA receptor. The authors note further that this state most closely corresponds to the state adopted by the NMDA receptor when bound to the negative allosteric modulator, ifenprodil. The authors however do not attempt to link these two findings. For example, does pre-incubation with ifenprodil oppose the ability of IgG2 from inhibiting GluN2B-containing NMDA receptors or vice versa? Also, how do the different inactive state of the channel, proposed in this study and in previous studies, relate to the closed states measured by single channel analysis (Yuan et al, 2009 amongst others). Given the broad readership of the Journal, it would be important to establish or at least propose a link between the structural information with functional characterization of the receptor.

It is a good point. Ifenprodil and Fab2 both promote the closed GluN2B state, ifenprodil by directly “gluing” the dimer interface shut, while Fab2 uses a network of long-range interactions to stabilize the closed ATD state. The importance of the closed GluN2B ATD in determining how the binding site residues communicate with the GluN2B ATD bottom lobe was assessed by simulating the Fv-free NMDAR as a control. Upon computing optimal paths, we identified a set of paths similar to those determined for the Fv2-bound systems. This indicates that Fv2 stabilizes the network of contacts present in the closed ATD state. Experimentally, pre-incubation of the GluN1-2B NMDARs with ifenprodil would inhibit the channel activity and thus, will mask inhibition by IgG2.

We agree that linking the structure to the single channel behaviors such as multi-modal activation scheme is one of the important goals. The evidence that the field has at the moment is that the “active” conformation can be stabilized by engineering of inter-subunit crosslinking by disulfide bonds in the LBD layer (Paoletti’s group in 2019) or by chemical cross-linking of ATDs

(Tajima 2016). While the field is working hard to correlate non-active1 and 2 to closed states in single channel recording, our group and others have not reached any conclusions, yet. Having said that, our patch clamp experiment shows that IgG2 favors desensitization (as measured by the macroscopic current) and therefore implies that the non-active1 conformation (favored by Fab2) likely represents desensitized state. This point is addressed in the revised manuscript.

Minor:

1. *The authors identify the different NMDA receptor subunits GluN1, GluN2A-D and GluN3A-B, however, they fail to define the splice variants, GluN1a and GluN1b. Some clarification is needed.*

We agree. The data we had in the original manuscript was on GluN1b-2B NMDAR that contained the exon5-encoded motif. We have now done electrophysiological recordings on the GluN1a-2B NMDAR and see inhibitory effect in both. Therefore, the effect appears to be independent of the GluN1 splicing. This is incorporated in the revised manuscript.

2. *There is no clarification in the main text for the abbreviations, Fv and Fab, which should be addressed.*

We agree. Thank you for pointing this out.

Antigen-binding fragment (Fab) and variable fragment (Fv) were addressed in the text.

Reviewer #5 (Remarks to the Author):

1. *The functional, structural, and molecular simulation results generally support and reinforce each other rather well. One place where the degree of complementarity falls short is the binding-site mutations. Here they made a speculation: "When the three residues were mutated to alanine, inhibition of IgG2 was mostly but not completely removed (Fig. 3E), perhaps indicating that the alanine mutations did not completely mask the binding capability..." Instead of speculation, the authors could carry out a binding assay of the mutants (as they did for wild-type in Fig. 2).*

We agree with the reviewer. As suggested, we have done the similar experiment as in Fig. 3E. We expressed GluN1-EGFP/GluN2B WT and mutants and ran FSEC (using GFP signal) in the presence and absence of IgG2. We see no binding with both the alanine and the tryptophan mutants. These results explain the decreased effects observed in the mutants. Unfortunately, with this method, we could not pick up some minor bindings that the alanine mutant may have. Nevertheless, this result is incorporated to the new Supplementary Fig. 5.

2. *Regarding Ig5, the authors made contradictory claims. In p. 5, the minor potentiating effect of IgG5 was touted as an example of upregulation. However, in p. 11, when explaining their cryo-EM data, they stated the data were consistent with little or no functional effects of IgG5.*

Thanks for pointing this out. We revised the manuscript accordingly. It now read: "...thus, is consistent with the observation that IgG5 does not mediate inhibition and instead has a small potentiating effect on the function of GluN1b-GluN2B NMDARs."

3. p. 10, "Although both IgG2 and IgG5 bind the GluN2B ATD, they do so at different protein surfaces (Fig. 4E-F)." It will be really helpful if the authors elaborate on these different binding surfaces.

The specific interactions for IgG2 and IgG5 are described in Fig. 3D and Fig. 4D, respectively. In Fig. 4E-F, we only wanted to show that the locations of the binding. We revised the text to clarify this. It now reads, "...Although both IgG2 and IgG5 bind the GluN2B ATD, they do so at different locations"

To examine the different binding surfaces, we computed the Fab binding site surface area for both Fabs using a method based on solvent-accessible surface area (SASA). Our findings are presented in the revised manuscript.

In addition to this additional analysis of the Fab-ATD binding surface, we performed simulations of an alternative conformation of the ATD-LBD linker to better study the ATD-LBD interface. Analysis of this new dataset is featured in Fig S9B.

Minor: p. 9, "Fig. 3E-H" -- should H be F?

p. 12, "First, application ... show" -- subject-verb mismatch.

Thank you for noticing our mistakes. They are fixed in the revised manuscript.

REVIEWERS' COMMENTS

Reviewer #1 (Remarks to the Author):

The authors have made numerous modifications to the manuscript to accommodate my earlier concerns.

The manuscript is well-written and clarifies many aspects that were ambiguous in the initial submission. It will have a significant impact on the field and will provide novel insights into how antibodies can interact with NMDA receptors and alter their function.

I have no additional comments.

Reviewer #2 (Remarks to the Author):

I am fully satisfied with the revision. I think the study is appropriate for publication in Nature communications.

One minor request, and I do not need to see a revised version: there are a few large clashes (>1 Å) in the pdb validation report for the first structure; please take a look at these and make sure they are warranted by the data.

Reviewer #5 (Remarks to the Author):

This revision has addressed all my concerns.

Response to Reviewer #2

I am fully satisfied with the revision. I think the study is appropriate for publication in Nature communications. One minor request, and I do not need to see a revised version: there are a few large clashes (>1 Å) in the pdb validation report for the first structure; please take a look at these and make sure they are warranted by the data.

We went over the structure, fixed the problematic region. This moderately corrected coordinate was submitted to the Protein Data Bank for eventual publication. These minor changes would not change scientific interpretation.